# Lost in Latent Space:
# Examining failures of disentangled models at combinatorial generalisation

**Milton L. Montero**[1,2][*]   **Jeffrey S. Bowers**[1]   **Rui Pont Costa**[2]
**Casimir J.H. Ludwig**[1]   **Gaurav Malhotra**[1]
[1]School of Psychological Science   [2]Department of Compute Science
University of Bristol
{m.lleramontero,j.bowers,rui.costa,c.ludwig,gaurav.malhotra}@bristol.ac.uk

## Abstract

Recent research has shown that generative models with highly disentangled representations fail to generalise to unseen combination of generative factor values. These findings contradict earlier research which showed improved performance in out-of-training distribution settings when compared to entangled representations. Additionally, it is not clear if the reported failures are due to (a) encoders failing to map novel combinations to the proper regions of the latent space, or (b) novel combinations being mapped correctly but the decoder being unable to render the correct output for the unseen combinations. We investigate these alternatives by testing several models on a range of datasets and training settings. We find that (i) when models fail, their encoders also fail to map unseen combinations to correct regions of the latent space and (ii) when models succeed, it is either because the test conditions do not exclude enough examples, or because excluded cases involve combinations of object properties with its shape. We argue that to generalise properly, models not only need to capture factors of variation, but also understand how to invert the process that causes the visual input.

## 1   Introduction

Disentangled representations extract factors of variations from data, and learning them has been the focus of much research in recent years [1]. Several approaches have been proposed to induce disentanglement, including latent space penalization [1–3], different training regimes [4, 5], architectures [6], data-driven inductive biases [7] and model selection methods [8]. These have produced more interpretable representations [9] that improve sample efficiency and learning for downstream models [10, 11, 8].

Importantly, researchers have also hypothesized that disentangled representations could provide a way of improving generalisation performance by enabling the discovery of causal variables in data [12] or by capturing its compositional structure [8]. This claim is especially interesting as it allows machine learning systems to emulate a key property of human intelligence – the ability to generalise to unseen combinations of known elements. For example, if a model has learned to generate red triangles and blue squares, then the model should also be able to correctly generate blue triangles and red squares. This property, which we refer to as *combinatorial generalisation*, gives humans the ability to make "infinite use of finite means" [13–16] and has been termed "a top priority for AI to achieve human-like abilities" [17]. Indeed, several authors have reported that unsupervised models that are better at disentangling generative factors are also better at some forms of combinatorial generalisation [1, 10, 6]

---

[*]Corresponding author

36th Conference on Neural Information Processing Systems (NeurIPS 2022).

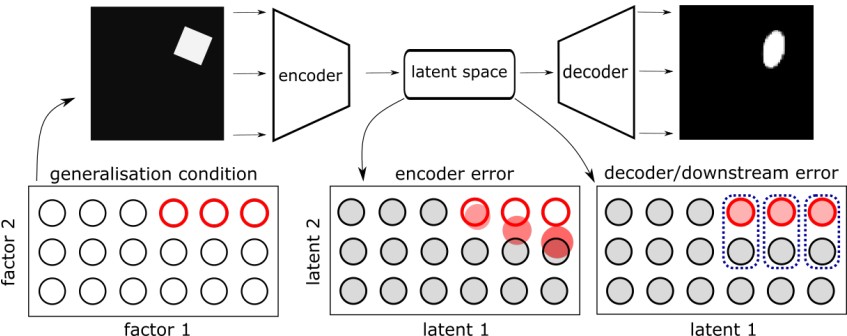

Figure 1: **Possible problems for combinatorial generalisation**. *Left*: visualisation of how a combinatorial generalisation condition can be defined for a dataset containing two generative factors. Black and red circles denote training and test examples, respectively. In the Middle and Right panels, the outline of each circle shows the target position of the representation in latent space, while a shaded circle shows the position at which the latent representation is projected by the encoder. *Middle*: first type of combinatorial generalisation error, encoder error. The encoder projects unseen inputs to different parts of the latent space than what was expected based on their generative factor values (shown as shaded circles falling outside their target outlines) *Right*: second type of combinatorial generalisation error, decoder error. Observed representations (shaded circles), are mapped to the correct position in latent space (circle outlines), but a decoder/downstream process mixes-up the black and red representations (blue dashed lines).

However, two recent studies have reported evidence that contradicts this hypothesis. Montero et al. [18] considered datasets where inputs varied along several dimensions (generative factors) and divided generalisation conditions into different types. They found that VAEs with highly disentangled latent representations succeeded at the easiest generalisation conditions where only one combination of all generative factors was excluded (a condition they termed *recombination-to-element*), but failed at more challenging conditions, where all combinations of a subset of generative factors were excluded (termed *recombination-to-range* by Montero et al. [18]). In another second study, Schott et al. [19] tested 17 learning approaches, expanding the analysis to other paradigms, architectures and task structures. They observed that models showed moderate success at generalising on some artificial datasets, such as 3DShapes. However, their ability to generalise dropped significantly on more realistic datasets and in more challenging generalisation conditions.

These contradictory results could be down to a number of differences between studies that report successes and those that report failures. For example, there were qualitative differences in the datasets that were tested and how the test conditions were set up. This is indeed what Montero et al. [18] conclude, writing that "it is not clear what exactly was excluded while training [previous models]", suggesting that previously observed successes may have been on the simplest generalisation condition where only very few combinations were left out. But this still leaves open the question of why models that achieve high degree of disentanglement nevertheless fail at combinatorial generalisation under more challenging conditions.

We see two possible reasons why this could happen. One possibility is that models correctly infer the values of latent variables (Fig. 1, left) for novel combinations of generative factors, but the decoder fails to map these unseen latent values to the (output) image space – *decoder error* (Fig. 1, right). Some evidence supporting this hypothesis was observed by Watters et al. [6], who looked at not only the output image reconstructions, but also at latent representations in a simple reconstruction task. Watters et al. [6] observed that, in their task, models that have more disentangled latent representations do indeed map unseen combinations of generative factors to the correct values in the latent space. But they only tested a very simple dataset, involving only two generative factors. Montero et al. [18] showed that models can sometimes succeed to generalise in simple settings where they can solve the combinatorial generalisation problem through interpolation, but struggle at more challenging settings where larger number of combinations are excluded from the training set.

Another possibility is that current models fail at harder forms of combinatorial generalisation (such as recombination-to-range) due to an encoding failure – *encoder error*. That is, the encoder fails to map these harder unseen combinations of generative factors to the correct values of variables in the latent space (Fig. 1, middle). If this were the case, then it reflects a more fundamental limitation of

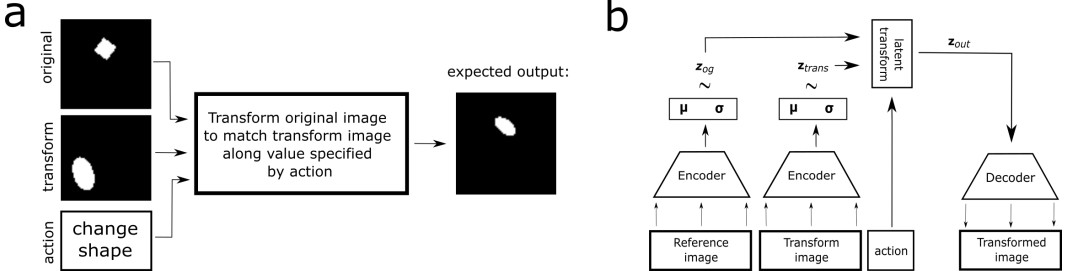

Figure 2: (a) **Image composition task**. An example of the composition task. In this case, the `[shape]` of the output must match the *transform* image and the rest of the values (`[position, orientation]`) must match the *original* image. (b) **Model architecture**. The model uses the same encoder to represent both images. Then a transform takes these latent representation and combines them to produce a transformed representation, which is used to reconstruct the output image. Reproduced from Montero et al. with permission from the authors.

current models, showing that even though models can map observed combinations to correct values of latent representations, they do not understand how generative factors are combined. So when novel combinations are presented, models cannot infer the values of generative factors that led to the observed data.

In this work we explore these issues by examining the latent space of generative models. Extending the findings of Montero et al. [18] and Schott et al. [19], we show that failure to generalise in the output space is accompanied by failure to generalise in latent space across several datasets. We show that these results hold for a broad range of decoder architectures (the standard *deconvolution decoder* and the *spatial broadcast decoder* [6]), loss functions ($\beta$-*VAE* and *WAE*) and task settings. Finally, by looking at the latent space we discover that, in addition to the difficulty of the generalisation task, a crucial condition for failure of combinatorial generalisation is the way in which generative factors are combined. These results not only challenge research that argues that disentanglement leads to better generalisation [10, 6], but also shows that the failure to generalise is a lot more entrenched in current models than suspected by Montero et al. [18] and Schott et al. [19].

## 2   Testing generalisation in latent representations

In this section we first describe a carefully designed semi-supervised task that enables us to achieve latent representations that are highly disentangled. This is crucial because standard disentanglement learning objectives, which rely on minimizing total correlation [1–3], struggle to reliably achieve a high degree of disentanglement, which makes it difficult to assess whether test data are projected to the expected position in the latent space (see Figure 1). Secondly, we describe our choice of corresponding model architectures as well as loss functions. For our results to be broadly applicable to existing research, we chose standard encoder and decoder architectures as well as a more sophisticated approach designed to induce a high degree of disentanglement [6]. Lastly, we describe some challenging test conditions for combinatorial generalisation on well known datasets.

### 2.1   Experimental Setup

#### 2.1.1   Task

One of the main barriers to achieving disentanglement in standard unsupervised training is the non-identifiability of the models when using iid data [20, 7]. In short, this is because there are infinitely many linear combinations of the underlying generative factors that produce a valid basis on which they can be represented.

To get around this problem we used the composition task developed by Montero et al. [18] (see Appendix A.1 for further discussion). This task takes two images and a query vector as inputs (see Figure 2(a)). The goal of the task is to output an image that combines the two images based on an *action* in the query vector:

$$\mathbf{input} = \mathbf{x_{og}}, \ \mathbf{x_{trans}}, \ \mathbf{q}$$
$$\mathbf{output} = \mathbf{x_{out}}$$

where $\mathbf{x_{og}}$ and $\mathbf{x_{trans}}$ are the two input images, $\mathbf{x_{out}}$ is the output image, and $\mathbf{q}$ is the query vector. Following Montero et al., we used actions that involve replacing one of the properties (generative factors) of $\mathbf{x_{og}}$ with a property of $\mathbf{x_{trans}}$, based on the value of $\mathbf{q}$. This vector was a one-hot encoding of the generative factor that was required to be changed.

By combining a semi-supervised component (the query vector plus image reconstruction) with sparse transitions (only one generative factor needed to be modified at a time), the composition task provided a strong inductive bias towards high levels of disentanglement, as has been shown in previous work that exploits these approaches [4, 5, 7].

### 2.1.2 Models

The model architecture used to solve the composition task is shown in Figure 2(b). It involves two encoders that map each image to a latent space. The latent vectors are then "composed" (see below), based on the query (action) vector. This composed vector is mapped from the latent space into the (output) image space using a decoder. Encoders and decoders for generative models match the ones used in past work [1, 3] for both `dSprites` and `3DShapes` datasets. For `MPI3D` we added another 2 more layers, which we found was necessary for models to learn the task. For the Circles dataset we used the encoder defined in Watters et al, along with the corresponding Spatial Broadcast Decoder architecture. Latent representations for the generative models are standard diagonal Gaussians with input dependent means and variances, to which we applied both the Variational and Wasserstein loss [21, 22].

We can define the composition operation that combines both latent representations as:

$$\mathbf{z_{out}} = \mathbf{z_{og}} \odot (\mathbf{1} - \mathbf{c}) + \mathbf{z_{trans}} \odot \mathbf{c}$$

where $\mathbf{z_{out}}$, $\mathbf{z_{og}}$ and $\mathbf{z_{trans}}$ are the latent representations corresponding to the output, original, and transform images and $\odot$ is the element-wise product between vectors. The variable $\mathbf{c}$ is the interpolation vector of coefficients. We defined two ways of computing $\mathbf{c}$: (i) a learned interpolation function computed as $\sigma(W \cdot [\mathbf{z_{og}}; \mathbf{z_{trans}}; \mathbf{q}] + \mathbf{b})$ with parameters $W$ and $\mathbf{b}$ where $[.;.]$ represents the concatenation of vectors; and (ii) a fixed interpolation function $\mathbf{c} = \text{padded}(\mathbf{q})$ which just pads $\mathbf{q}$ with zeros to match the vector length of the latent representation (see Appendix A.2 for more details).

To facilitate our goal of testing combinatorial generalisation, we augmented the target output by requiring the model to reconstruct the input images as well. Thus the trained encoder and decoder form a valid standard autoencoder. We can then probe the image reconstructions and the latent representations for unseen combinations to check if the models generalise. An illustration of this model can be found in Figure 2.

### 2.1.3 Datasets

We tested our models on three standard datasets: `dSprites` (a sythesised dataset of 2D shapes procedurally generated from 5 independent generative factors [23]), `3DShapes` (another synthesised dataset consisting of 3D shapes procedurally generated from 6 ground truth generative factors [24]) and `MPI3D` (a collection of four different datasets consisting of synthetic as well as real-world images procedurally generated from 7 generative factors [25]). These are the most popular datasets used in the disentanglement literature that also provide ground truth generative factor values. We tested combinatorial generalisation by systematically excluding combinations of values of generative factors from each dataset. Here we focus on the conditions described as *recombination-to-range* by Montero et al.. These are the most interesting conditions where, according to Montero et al., one would expect a model that learns disentangled representations to succeed at combinatorial generalisation, but tested models typically fail. Consider a dataset with, say, four generative factors [$g_1$, $g_2$, $g_3$, $g_4$] where all $g_i \in [0, 1]$. The *recombination-to-range* condition creates a training/test split where all examples with combinations of a subset of generative factors are excluded from the training set and added to the test set. Thus, an example of a dataset that tests recombination-to-range may consist of a training set where all combinations where [$g_1 > 0.5$, $g_2 > 0.5$] have been excluded from the training set and added to the test set. Note that the model trained on such a datasets would come across a number of

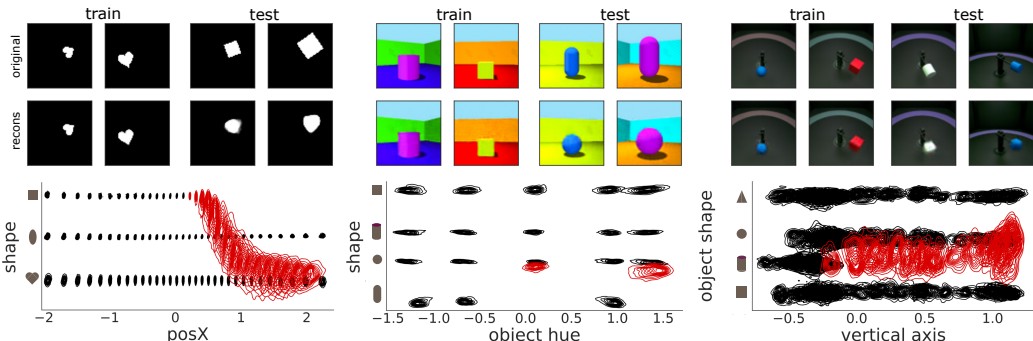

Figure 3: **Generalisation in latent space** *Top*: Typical examples of reconstructions for training and test images for models that learn highly disentangled representations. *Bottom*: Visualisation of the latent space of models trained on three datasets – `dSprites` (left), `3DShapes` (middle) and `MPI3D` (right). Each panel shows contours from the joint distribution of two latent variables that best predict the corresponding generative factors. In all cases, the red distributions indicate test data and the black ones indicate training data. Note that the latent values in this figure are *not* necessarily the same as the values of generative factors excluded. This is because every model ends up with a different internal representation based on its initialisation and sequence of training trials. However, in each case, we observed that these internal representations were highly structured when the model showed a high degree of disentanglement.

examples where $[g_1 > 0.5]$ and also examples where $[g_2 > 0.5]$, but never be trained on an example where both these conditions are true simultaneously. This method was used to create training / test sets for each of the datsets in the following manner:

- `dSprites`: All images such that `[shape=square, posX> 0.5]` were excluded from the training set. Squares never appear on the right side of the image, but do appear on the left and other shapes (hearts, ellipsis) are observed on the right as well.
- `3DShapes`: All images such that `[shape=pill, object-hue=> 0.5]` were excluded from the training set. Thus, pills colored as any of the colors in the second half of the HSV spectrum did not appear in the training set. These colors (shades of blue, purple, etc) were observed on the other shapes, and the pill was observed with other colors such as red and orange.
- `MPI3D`: All images such that `[shape=cylinder, vertical axis> 0.5]` were excluded from the training set. We also excluded all images where `[shape=cone]` or `[shape=hexagonal]` as these shapes are very similar to the pyramid and cylinder, respectively and make it hard to access reconstruction accuracy.

### 2.1.4 Measuring Disentanglement

We used the DCI metric from Eastwood and Williams [26] to measure the degree of disentanglement (see Appendix A.5). This metric provides a set of regression weights between ground truth factors and latent variables. We selected the latent variable corresponding to each ground truth factor using a matching algorithm based on these weights. Crucially, for highly disentangled models any combination of generative factors will have a corresponding combination of latent variables of the same cardinality. This allows us to create visualisations of the latent representations analogous to the ones in Figure 1.

## 2.2 Results

We observed that most models trained on the composition task managed to achieve a reasonably high degree of disentanglement, and successfully reconstructed images in the training data (see Appendix B.1 and B.2 for quantitative results). Here we present results of typical models that achieved a very high degree of disentanglement (with alignment scores typically > 0.95, see Appendix B.1 for details). Figure 3 shows reconstructions as well latent space representations of three typical models trained on the `dSprites`, `3DShapes` and `MPI3D` datasets, respectively.

Replicating results of Montero et al. [18], we observed that models showed poor generalisation to unseen combinations for all three datasets (see reconstruction of test images in Figure 3). For

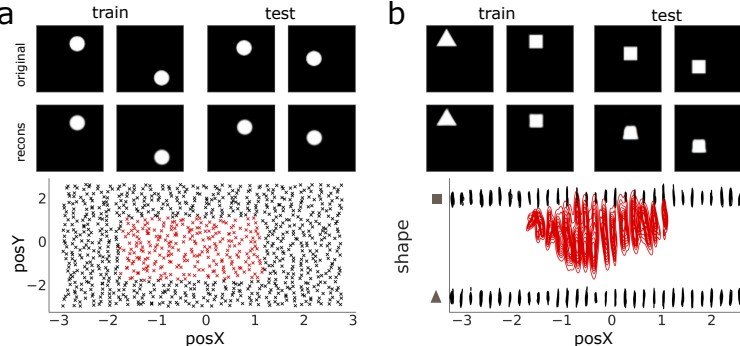

Figure 4: **Latent space induced by SBD** Visualisation of the latent space for models that use a Spatial Broadcast Decoder as described in Watters et al. Both visualisations correspond to a condition where combinations in the middle of the image are removed. (a) The result when this is done for the Circles dataset which uses only one shape and where the training set excludes all combinations `[shape=circle, 0.35 < posX < 0.65, 0.35 < posY < 0.65]` (b) The result for the a similar condition for the Simple dataset which uses two shapes and excludes all combinations `[shape=square, 0.35 < posX < 0.65, 0.35 < posY < 0.65]`.

`dSprites`, the models typically produced images in which the target location is correct but the shape is not (e.g., replacing the square with an ellipse). Similarly, with `3DShapes` and `MPI3D`, models typically reconstructed shapes with the correct colour hue (`3DShapes`) or vertical location (`MPI3D`), but made an error in the shape of the target object (e.g., replacing the pill (`3DShapes`) or cylinder (`MPI3D`) with a sphere(`3DShapes`) or pyramid (`MPI3D`).

Next, we looked at the latent representations of each of these cases by selecting a latent variable that showed highest correlation with each generative factor as described in Section 2.1.4. Figure 3 plots the distributions of the value of each latent variable for every combination of values of the relevant generative factors (see Appendix A.5 for details). Visualising these probability distributions for the combination of generative factors seen during training (in black) confirmed that the models learned highly disentangled representations (note the low variance of the probability distributions for trained combinations, especially for `dSprites` and `3DShapes`).

Crucially, we observed that the encoder failed to map the generative factors for the test combinations to the correct location in the latent space (red distributions). This was particularly true for `dSprites` and `3DShapes` dataset: note how the mean of the probability distributions for the left out combinations (`[shape=square, posX> 0.5]` for dSprites and `[shape=pill, object-hue=[blue, purple]]`) are shifted to overlap the latent representations seen during training. The shifts in the mean of distributions were less acute for `MPI3D`, but we nevertheless observed an increase in the variability of test distributions and a consistent shift in the location for values of generative factors that were far from the trained values.

In summary, we made three key observations in these experiments. First, models replicated the negative results reported previously: i.e., a degradation in reconstruction performance in the challenging combinatorial generalisation conditions. Second, by visualising the latent space, we observed that failures of generalisation coincided with poor latent representations, showing that failures of generalisation are not entirely due to decoder errors. Third, the failures in latent representations as well as output reconstructions showed that different generative factors have important qualitative differences, with models making larger errors in generating the correct shape for unseen combinations than in reproducing the position, scale or color (see Appendix B.2).

## 2.3 Exploring alternative hypotheses

In the experiments above, we used the deconvolution network as the decoder, which is the standard decoder used in VAEs [1]. However, some studies have shown that replacing the deconvolution network with a different architecture helps the model learn representations that are more disentangled. One such architecture is the Spatial Broadcast Decoder (SBD from hereon) developed by Watters

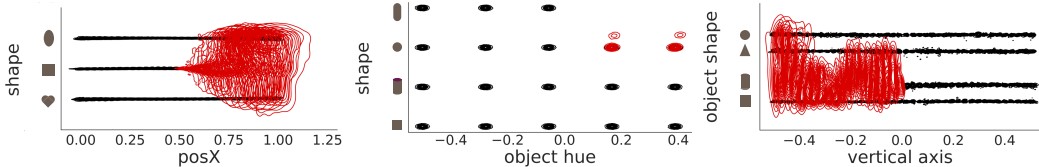

Figure 5: **Replacing decoder with ground-truth** Visualisation of the latent space for models trained to predict the correct generative factor values for each image. The visualizations correspond to the same test conditions as the ones presented in Figure 3: (a) `dSprites`, where all combinations where `[shape=square, posX > 0.5]` were excluded from the training set (b) `3DShapes`, where all combinations where `[shape=pill, object-hue > 0.5]` were excluded, and (c) `MPI3D`, where all combinations where `[shape=cylinder, vertical-axis > 0.5]` were excluded (note that, in this case, `[vertical-axis > 0.5]` corresponds to the corresponding latent variable value $< 0$).

et al. [6], who showed that such a decoder not only helped in learning representations that were highly disentangled, but also helped in solving the problem of combinatorial generalisation.

In our next set of experiments, we tested whether using the SBD succeeds at combinatorial generalisation in more challenging settings, akin to the ones we tested above. We first replicated the results for both conditions tested by Watters et al. [6], using our composition task on the Circles dataset and replacing the deconvolution network with the SBD. Figure 4 (panel A) shows the results for the first condition, where the training set excluded images with circles presented in the middle of the canvas (see Appendix B.2 for results on the second condition). As we can see from this figure, the latent space learned by the model was indeed highly structured and, crucially, the model mapped the latent values for the left out combinations (red crosses) to the correct region of the latent space.

Next, we wanted to check how these results scaled when we made the generalisation test more challenging, excluding a combination of values for a range of variables (the *recombination-to-range* condition developed by Montero et al. [18]). We can do this with the Circles dataset by adding a third generative factor and the simplest way of doing this is by letting the shape take one of several values, instead of being a circle for all examples.

We tested this by constructing a new dataset called Simple. Like the Circles dataset, this dataset consisted of images that had a shape located at various (x and y) positions on the canvas. Unlike the Circles dataset, this shape could be either a triangle or a square (we chose these shapes as they are different enough for models to not confuse them by accident). We then tested the two conditions similar to the conditions tested by Watters et al. [6], excluding all combinations where one of the shapes was presented in the middle (first condition) or the top-right (second condition) of the canvas.

The results for the first condition are shown in panel B of Figure 4 (see Appendix for results of the second condition). Like the results for the first set of experiments above, we have plotted the probability distribution of combinations of values of the the latent variables (here, shape and posX) for all values of the third latent variable (here, posY). Similar to our earlier results, we observed that even the model using the SBD failed to map the unseen (test) combinations to the correct position in the latent space (compare the mean and variability of the red distributions compared to the black ones). Correspondingly, we also observed that models failed to correctly reconstruct output images for the test conditions, even when they had no problem reconstructing these images for combination of generative factors seen during training (See Appendix B.2).

## 3 Understanding the role of the encoder

We have shown that errors in generalisation occur in latent space as well in the reconstructions for generative models. To conclude this analysis we show that the problem does not lie in the generative nature of the task by testing the encoders on a latent prediction task, where the target is not the ground truth factor values. We used the same datasets and generalisation conditions as before.

### 3.1 Experimental setup

We trained models on a prediction task where the model was given a single image as the input and the target was to output the ground truth generative factor values. This ensures that the resulting latent representation will be completely disentangled, as the model must output each of the factor

values separately. We used the mean squared error of the output and target vectors as a learning signal. All other parameters remain same as the composition task. We used the same architectures for the feed-forward models as the encoders in the corresponding datasets above.

We evaluated the models using the $R^2$ metric as used in Schott et al. [19] to check whether the models had solved the task. We also use the same method to visualise the results as described above, plotting the joint probability distributions for the combination of generative factors along which the model was tested for generalisation. In this case, it is straightforward to do so as the relevant output dimensions shared the same index as the corresponding generative factor.

Because a supervised learning setting might not give such a rich signal to the model, we also tested a different scheme that still tests the encoder. In this alternative setting, we first train a complete generative model on the full data set to a high degree of disentanglement. We then freeze the decoder and *retrain the encoder* on the generalisation conditions we are interested in (see Figure 33 in Appendix D.2). Thus, the encoder has to reproduce the latent representations that the decoder is expecting in order to properly reconstruct unseen samples. Simultaneously, errors can only be a product of failures by the encoder, as the decoder is completely disentangled during its training.

## 3.2 Results

The results for the three dataset are shown in Figure 5. We observed a very similar pattern for this task as the semi-supervised task above. For all three datasets, the probability distributions of latent values for the left out combinations (in red) showed a much higher variance and were shifted towards combinations of latent values that had been experienced during training. Thus, even when the encoders were trained to recognise perfectly disentangled generative factors, they failed to generalise to combinations that were not experienced during training. For the frozen decoder setting, we include results in Appendix D.2, showing that the failures occur there as well.

## 4 Explaining contradictory findings

How can we reconcile the failures of generalisation observed above with with past studies that showed successful combinatorial generalisation on some datasets and conditions? One response is that the generalisation conditions we test are more challenging than the ones tested in previous studies because they exclude more data from the training set. However, it is also possible that the success of combinatorial generalisation is not only determined by how many combinations are excluded from the test condition, but also by which combination of generative factors are tested.

All failures observed above were for cases where the generative factor 'shape' is combined with another generative factor (position / color). It is possible that this was because shape combines with other factors in a qualitatively different manner than, say, the floor hue combines with the wall hue in `3DShapes`. In order to completely disentangle shape from, say, position the model must come up with a representation of shape that is invariant to a change in position. And it must do so by learning from training examples where shape and position jointly determine the same part of the image.

We illustrate these two types of combinations in Figure 6. In the first case ( e.g., `[floor-hue, wall-hue]`), two factors determine mutually exclusive pixels of the image. We call these types of combinations *non-interactive*. Solving the combinatorial generalisation problem in the non-interactive case is trivial – the model can do this by learning independent mappings between sets of pixels and the corresponding generative factors. In the second case (e.g., `[shape, posX]`), the value of pixels is jointly determined by two factors in a nonlinear and complex manner. A pixel node may have value $+1$ for a given value of `[shape]` and `[posX]`, but this value may change to $-1$ for a slight change in either factor. Other changes will have no effect on the value of some pixels. We call these types of combinations *interactive*. To succeed at combinatorial generalisation in the interactive condition, a model must not only learn the mapping between pixels and factors, but also learn how the factors combine to jointly determine the value of each pixel.

Based on this insight, we hypothesised that models may fail to perform combinatorial generalisation in the interactive condition, but succeed in the non-interactive condition. To test this hypothesis, we carried out a set of experiments where the model had to solve the problem of combinatorial generalisation, but the combinations that were excluded did not interact in the training examples. We can do this for both the `3DShapes` dataset as well as the `MPI3D` dataset, as both datasets involve

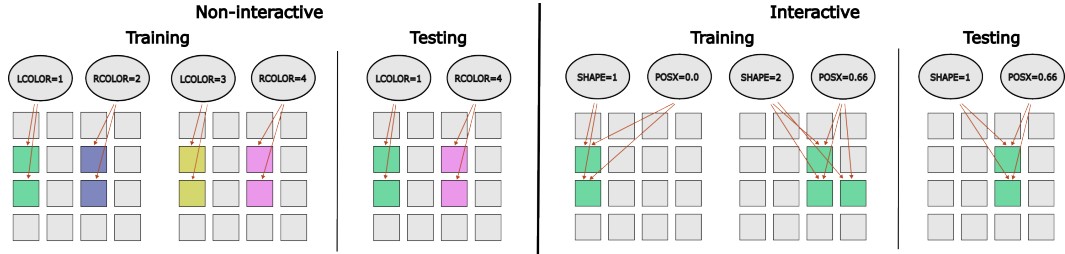

Figure 6: **Non-interactive and interactive combinations as graphs**. The panels on the left show an example of two factors `[lcolor, rcolor]` that we call non-interactive. In this case, the value of each factor determines the colour of mutually-exclusive set of pixels. The model must learn the dependencies (edges) between the generative factors and pixels. Compare this to the three panels on the right, illustrating the interactive condition. In this case, the two factors `[shape, posX]` jointly determine the value of each pixel. In this case, the model must learn how the value of shape nodes non-linearly determines the value of pixel nodes based on the value of posX node.

images where the canvas contains several disjoint elements. So, we repeated the semi-supervised learning experiments above, where models had to learn the composition task, but instead of excluding combinations where the generative factors interacted with each other, we excluded the following combinations from the training set:

- 3DShapes: Exclude all combinations such that `[floor-hue < 0.25, wall hue > 0.75]`: none of the training images had the combination of floors with a "warm" hue and walls with a "cold" hue.
- MPI3D: Exclude all combinations such that `[shape={cylinder,sphere}, background color=salmon]`. Note, even though shape combinations are excluded, the combination of generative factors excluded do not interact (i.e. determine different parts of the image).

It is also possible that model struggled with factors that combine with `[shape]` not because it is an interactive factor, but because it is a 'discrete' factor, taking on a few specific and unrelated values. If this is the case, then a model should be able to succeed at combinatorial generalisation by learning discrete latent representations. A recent model – CascadeVAE [27] – addresses this by concurrently inferring continuous and discrete latent factors. We tested this model on the image reconstruction task in the recombination-to-range condition of dSprites dataset.

Finally, it is possible that standard VAEs struggle with the combinatorial generalisation because they cannot capture the dependencies between generative factors. A recent model – Commutative LieGroupVAE [28] – uses an adaptive equivariant structure, rather than a fixed vector space, to learn factors of variation in the data. This approach combines explicit modeling group operations plus penalties to the learned basis in order to learn a highly disentangled representation of the input (see the original work for more details). The hope is that because this method is adaptive, it may be able to capture not only the generative factors underlying the data, but also the dependencies between them (see Figure 6). As with CascadeVAE, we use the same training configuration as in the original work and test on the interactive conditions of dSprites and Shapes3D datasets (see details in Appendix D.4).

## 4.1 Results

For the non-interactive conditons, we again observed that models learned highly disentangled representations for this task. More interestingly, we also observed that models now succeeded at the combinatorial generalisation task. Figure 7 shows some typical reconstructions for training as well as test images. These reconstructions successfully reproduce the unseen combination of floor hues and wall hues for the 3DShapes dataset and the combination of shape and background arc color for the MPI3D dataset. This figure also shows the joint probability distribution for trained as well as novel combination of latent values. In contrast to Figure 3, we now see that the means of the test latent distributions (in red) fall in the expected location and a show a much smaller variance.

We also observed that alternative models cannot solve the hard generalisation challenges described (See Appendix D). Figure 35 shows these results for CascadeVAE on dSprites (the only dataset for which authors provide training parameters), and how the model clearly fails at combinatorial generalisation as before, unable to map the previously seen shape to a novel position. Figure 37

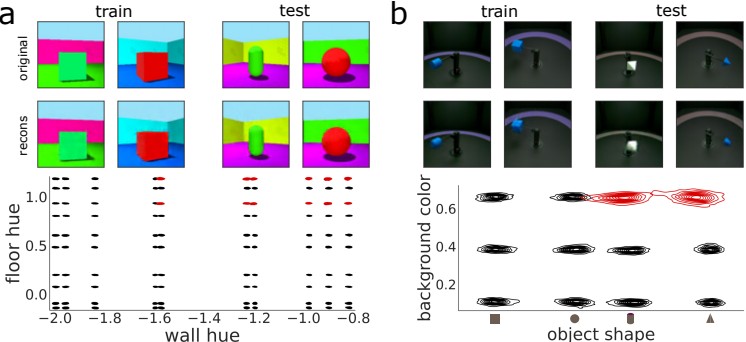

Figure 7: **Latent space for conditions where models succeed** Visualisation of the latent space for models trained on conditions in which they succeed at generalisation. The figure follows the same conventions as in Figure 3. (a) Results for 3DShapes when removing floor and wall color combinations. (b) Visualisation for the MPI3D dataset for combinations where the salmon background has not been seen with the cylinder shape.

shows the same results for LieGroupVAE, with model succeeding at the non-interactive condition in Shapes3D but failing the interactive one.

## 5  Discussion

Our world is inherently compositional – it can be decomposed into simpler parts and relationships between these parts. The idea behind learning disentangled representations is to recover this underlying compositional structure of the world from perceptual inputs. Unsupervised learning models, such as VAEs, aim to do this by separating out the factors that remain invariant under transformations of other factors [29]. It is therefore tempting to conclude that models that manage to show a high degree of disentanglement on the training set also capture the compositional structure of the world. If they did, they should be able to generalise to settings that present novel combination of the factors of variation. However, in our experiments, we observed that models that learned highly disentangled representations nevertheless failed at combinatorial generalisation, not only in the reconstruction space, but also in the latent space. These failures reproduced over several different datasets, different types of decoders, different loss functions and under a variety of different task settings.

Our interpretation of these results is that models manage to achieve a high degree of disentanglement by discovering factors that remain invariant over training examples and simply associating perceptual inputs with these factors. However, to capture the compositional structure of the world, models must additionally understand how factors interact to cause the perceptual input – that is, develop a good causal model of the world. We think the failure of models to form such a causal model is the reason why models succeeded at problems of combinatorial generalisation that do not involve an *interaction* between the left-out factors (Figure 7), but consistently failed at problems where these factors interacted (Figure 3). When generative factors do not interact, the model does not need to learn how the factors combine to determine the same part of the output space. Instead, it can simply map the value of each generative factor to a different location. It *appears* to solve combinatorial generalisation, but it does not understand how generative factors combine.

In summary, our work shows that the problem of combinatorial generalisation remains unsolved in both latent space and reconstruction space, even for highly disentangled models. By highlighting this limitation, we hope to inspire more work exploring new approaches that emulate this key capacity of human cognition, thus endowing models with a better understanding of the compositional structure of our world.

## Acknowledgments

The authors would like to thank the members of the Mind & Machine Learning Group for useful comments throughout the different stages of this research. This research was supported by a ERC Advanced Grant, Generalization in Mind and Machine #741134.

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
