# Supplementary Material for Lost in Latent Space

## A  Experimental setup

### A.1  Composition task

Here we describe the image composition task in more detail. This task was created by Montero et al. [18] to check whether a different task than simple image reconstruction helps the model to learn more disentangled representations and as a consequence show combinatorial generalisation. At a high level the task is a manipulation task, where models must learn to extract the relevant factors $g_i$ from two images, $X_{og}$ and $X_{trans}$, and combine them based on a given 'action':

$$\textbf{input} = X_{og},\ X_{trans},\ \mathbf{q}$$
$$\textbf{output} = X_{out}$$

where $X_{out}$ is the output image and $\mathbf{q}$ is the query vector, that encodes the action. Following Montero et al., we use actions that involve replacing one of the properties (generative factors) of $X_{og}$ with a property of $X_{trans}$. The query vector uses a one-hot encoding of the generative factor that must be changed. See Figure 2 for an example of this task.

A critical aspect of this task is that it requires the change to be only one generative factor for each trial. Restricting the manipulation to a single generative factor encourages sparseness of representations and provides a strong inductive bias for the model to learn disentangled representations. Previous research has shown that sparse transitions following natural image statistics provide a strong inductive bias towards disentanglement (see, for example, Klindt et al. [7]). This is because breaking the i.i.d. assumption, present in most datasets, allows for the identification of the underlying factors (see Klindt et al. [7] for details and Hyvarinen and Morioka [20] for a similar argument for selecting independent components from data).

A second inductive bias towards disentanglement is provided by the nature of the composition task. In contrast to a reconstruction task, that doesn't provide any supervision signal, and a classification task, which provides a strong supervision signal, the composition task uses a query vector to provide a *weak supervision* signal. Two studies have shown that a weak supervision signal can provide a strong inductive bias for generalisation. Locatello et al. [4] showed that a few labeled examples were enough to significantly improve the disentanglement in generative models. In the second study, Lin et al. [5] developed a task with a weak supervision signal, where the target was to identify the factor that had changed between two images. They used this task to train a GAN and show it leads to more disentangled representations. The query vector in the composition task constitutes a similar weak supervision signal, but provided as an input instead of as a target.

### A.1.1  Procedure

When training models, we sampled each combination in an online fashion. The procedure works as follows:

1. Sample an image $X_{og}$ and with generative factors given by the vector $\mathbf{g_{og}}$ from the dataset.
2. Sample an action, $a$, that indexes the set of all generative factors, $\{g_1, \ldots, g_n\}$.
3. Sample a second image $X_{trans}$ such that $\mathbf{g_{trans}}[a]$, the $a$th generative factor of $X_{trans}$ does not match $\mathbf{g_{og}}[a]$, the $a$th generative factor of $X_{og}$.
4. Compute $\mathbf{g_{out}}$ by replacing $\mathbf{g_{og}}[a]$ with $\mathbf{g_{trans}}[a]$ and get and it's associated image $X_{out}$.

All sampling is done uniformly and we allow sampling of categorical variables such as shape.

### A.2  Models

### A.2.1  Composition operation

We use the general architecture described in Figure 2(b) to define a model that solves the composition task. As mentioned in the main text this requires the definition of the composition operation, $f(\cdot)$ in latent space. In general, such operation can be defined as:

$$\mathbf{z_{out}} \triangleq f(\mathbf{z_{og}}, \mathbf{z_{trans}}, \mathbf{q})$$

where $\mathbf{z_{out}}$, $\mathbf{z_{og}}$ and $\mathbf{z_{trans}}$ are the latent representations corresponding to the output, original, and transformation images. and $\mathbf{q}$ is the one-hot vector encoding the action to be performed.

We defined 3 different versions of this operation:

$$f_{mlp}(\mathbf{z_{og}}, \mathbf{z_{trans}}, \mathbf{q}) = \mathbf{W}_{out} \cdot \text{ReLU}(\mathbf{W}[\mathbf{z_{og}}; \mathbf{z_{trans}}; \mathbf{q}] + \mathbf{b}) \tag{1}$$

$$f_{lin}(\mathbf{z_{og}}, \mathbf{z_{trans}}, \mathbf{q}) = W_{out} \cdot [\mathbf{z_{og}}; \mathbf{z_{trans}}; \mathbf{q}] \tag{2}$$

$$f_{interp}(\mathbf{z_{og}}, \mathbf{z_{trans}}, \mathbf{q}) = \mathbf{z_{og}} \odot (1 - \mathbf{c}) + \mathbf{z_{trans}} \odot \mathbf{c} \tag{3}$$

where $[.;.]$ represents the concatenation of vectors, $\mathbf{c}$ is the interpolation vector of coefficients. We define $\mathbf{c}$ in two ways for equation 3 as described in the main text:

$$\mathbf{c_{learn}} = \sigma(\mathbf{W} \cdot [\mathbf{z_{og}}; \mathbf{z_{trans}}; \mathbf{q}] + \mathbf{b}) \tag{4}$$

$$\mathbf{c_{action}} = \text{pad}(\mathbf{q}, \text{length}(\mathbf{z}) - \text{length}(\mathbf{q})) \tag{5}$$

where the first option has learnable parameters $W$ and $\mathbf{b}$. The second just pads the query vector, $\mathbf{q}$, with zeros to match the vector length of the latent representation.

Preliminary tests showed that using (1) or (2) as composition operations lead to poor disentanglement. Thus we only tested the two variants of (3) in the rest of the experiments. For some datasets such as dSprites, (5) worked better. For 3DShapes on the other hand, (4) gave better results.

For the rest of the architectures (encoders and decoders) we use the same parameters across datasets for each condition. For dSprites and 3DShapes, we use a similar architecture as in Kim and Mnih [3] but increase the number of channels in the early convolutions. MPI3D required us to increase these values again in order to achieve good reconstructions. For Circles and Simple we use the architecture proposed in Watters et al. [6] but slightly increase the channel size in the decoder to avoid a performance overhead related to PyTorch's implementation. See Table 1 for details.

Table 1: **Architecture parameters**

| dSprites & 3DShapes | | MPI3D | | Circles & Simple | |
|---|---|---|---|---|---|
| Encoder | Decoder | Encoder | Decoder | Encoder | Decoder |
| Conv(32, 4, 2) | Transpose | Conv(64, 4, 2) | Transpose | Conv(64, 4, 2) | SBD(64, 64) |
| Conv(32, 4, 2) | of encoder | Conv(64, 4, 2) | of encoder | Conv(64, 4, 2) | Conv(64, 5, 1) |
| Conv(64, 4, 2) | | Conv(128, 4, 2) | | Conv(64, 4, 2) | Conv(64, 5, 1) |
| Conv(64, 4, 2) | | Conv(128, 4, 2) | | Conv(64, 4, 2) | Conv(64, 5, 1) |
| Conv(128, 4, 2) | | Conv(256, 4, 2) | | Linear(256) | Conv(64, 5, 1) |
| Linear(256) | | Linear(256) | | Linear(20) | |
| Linear(20) | | Linear(20) | | | |

All layers where followed by ReLU activation functions except for the last one. The latent layers used were 10-dimensional, parameterized diagonal Gaussians in all cases. For both the Conv and DeConv layer the parameters indicate number of channels, size of convolution filter and the stride. "Same" padding was used throughout. For completion, we note that the architectures for the latent prediction task were the same ones as the corresponding encoder.

## A.3 Datasets

Throughout the article we use standard datasets used to test disentangled models in the literature. These are: dSprites, 3DShapes, MPI3D and Circels. Below we describe each of the generative factors they contain. We've focused on datasets that have been specifically designed for this purpose and thus contain explicit values of the generative factors. This is necessary in order to compute the level of disentanglement with most metrics, including the DCI metric.

- dSprites: Introduced in Higgins, Higgins et al. [1] to test the original $\beta$-VAE approach to disentanglement. It contains the following generative factors: `[shape, scale, orien tation, position X, position Y]`. Orientation here refers to the rotation of the shape along it's center of mass. This is as opposed to the meaning in `3DShapes` (see below). The GitHub repository for `dSprites` can be found at: https://github.com/deepmind/dsprites-dataset.

- `3DShapes`: Introduced in Kim Kim and Mnih [3], to study the FactorVAE penalty. It contains the 6 generative factors: `[floor hue, wall hue, object hue, scale, shape, orientation]`. Colors are defined in the HSV format, and the values correspond to the hue component. Here orientation defines the angle of point-of-view for the scene. The objects themselves do not rotate. This dataset can be found at: `https://github.com/deepmind/3d-shapes`.

- `MPI3D`: This dataset was proposed in Gondal et al, Gondal et al. [25] as part of the the NEURIPS Disentanglement Challenge. It contains seven generative factors: `[object color, object shape, object size, camera height, back ground color, horizontal axis, vertical axis]`. We note that vertical axis and horizontal axis have complex non-linear dependencies between them in the rendered image. The names can also be misleading as horizontal axis controls the height of the arm while vertical axis controls the rotation in the direction perpendicular to the horizontal axis. The GitHub repository for this dataset can be found at: `https://github.com/rr-learning/disentanglement_dataset`.

- `Circles`: Introduced to test the capabilities of the Spatial Broadcast Decoder models in Watters et al, Watters et al. [6]. It contains only two factors: `[position x, position y]`, the model being required to only render circles at the relevant position. There is no published dataset in this case, though the authors mention that they generated this dataset using the Spriteworld library, which is a library designed to generate simple datasets for reinforcement learning. We adapted the library to generate still images similar to those found in Watters et al. [6].

- `Simple`: We modified the Circles dataset to create the Simple Sprites dataset in order to test the effect of introducing an additional shape when testing the combinatorial generalisation capabilities of the Spatial Broadcast decoder. In addition to the two position factors in the Circles dataset, we added a shape factor that could take one of two values `[shape={triangle,square}]`. We chose these shapes so that they are sufficiently different from each other. To generate the dataset we use the same library that we use to replicate the Circles dataset above.

### A.3.1 Combinatorial generalisation test conditions

For each dataset we tested one success condition and one failure condition. These all constitute recombination-to-range conditions as definde in Montero et al. [18] but we refer to them as combinatorial generalisation conditions [2]. The failure conditions were defined as follows

- `dSprites`: All images such that `[shape=square, posX> 0.5]` were excluded from the training set. Squares never appear on the right side of the image, but do appear on the left and other shapes (hearts, ellipsis) are observed on the right as well.

- `3DShapes`: All images such that `[shape=pill, object-hue=> 0.5]` were excluded from the training set. Thus, pills colored as any of the colors in the second half of the HSV spectrum did not appear in the training set. These colors (shades of blue, purple, etc) were observed on the other shapes, and the pill was observed with other colors such as red and orange. Additionally we removed every other color from the dataset. This helped us clearly discretize the generative factors and latent representations, allowing us to clearly observe the performance of the models in the latent space.

- `MPI3D`: All images such that `[shape=cylinder, vertical axis> 0.5]` were excluded from the training set. We also excluded all images where `[shape=cone]` or `[shape=hexagonal]` as these shapes are very similar to the pyramid and cylinder, respectively and make it hard to access reconstruction accuracy. Furthermore, only images with `[horizontal axis=0]` were included as rotation of both horizontal and vertical axis make it hard to find completely unseen combinations due to how much rotation of the objects is involved.

- `Circles`: All images such that `[posX 0.5,posY> 0.5]` were excluded from the training set. Models have not seen circles in the top right corner.

---

[2]This is the terminology used Schott et al. [19], though, technically speaking, their interpolation condition was also a combinatorial generalisation case analogous to the recombination-to-element condition in Montero et al.

- `Simple`: All images such that $[0.35 < \text{posX } 0.65, 0.35 \text{ posY} > 0.65, \text{shape=triangle}]$ were excluded from the training set. Similar to the previous condition, but now the unseen triangles lie in a patch in the middle of the screen. Squares were presented on all positions, similar to dSprites.

And the success ones as:

- `3DShapes`: These test images presented a novel combination of generative factors, here $[\text{floor hue } < 0.25, \text{ wall hue } > 0.75]$ – that is, the model has seen all wall hues and floor hues in the range $[0, 1]$, but it has never seen a combination a floor with a hue $< 0.25$ with a wall of a hue $> 0.75$. In this case no discretization was necessary, as the task is easy enough that the model obtained good disentanglement and the visualisations were easy to understand.

- `MPI3D`: All images such that $[\text{shape=\{cylinder,sphere\}, background color=salmon}]$ were excluded from the training set. We also excluded all images where $[\text{shape=cone}]$ or $[\text{shape=hexagonal}]$ as these shapes are very similar to the pyramid and cylinder, respectively and make it hard to access reconstruction accuracy. Furthermore, only images with $[\text{horizontal axis=0}]$ were included as rotation of both horizontal violate our independence assumption. Models can appear over the background strip very often.

- `Circles`: All images such that $[0.35 < \text{posX } 0.65, 0.35 \text{ posY} > 0.65]$ were excluded from the training set. Models have not seen circles in a patch located in the middle of the image.

## A.4 Training

To penalise the models we use the standard VAE objective [21] and the Wasserstein loss (WAE, Tolstikhin et al. [22]). In the case of the MPI3D dataset, WAEs were the only one that could successfully converge for the semi-supervised setting, which we use as a sanity check. This is likely because the VAE objective forces the distribution of the latent representation for each input to match the prior. On the other hand WAEs only require the marginal posterior *over all inputs* to match the prior. The result is a latent representation that is more flexible, which seems to be required in this more complex dataset. As a consequence, we do not test VAE models on the MPI3D dataset for the composition task.

We trained the models to convergence, which in all cases occurred within 100 epochs. Batch sizes were kept at 64 and learning rates at $1e-4$ as in previous work for dSprites, 3DShapes and MPI3D. For the Circles dataset we used a batch size of 16 and a learning rate of $3e-4$ as in Watters et al.. We used the Adam optimizer with the default PyTorch values for all remaining parameters [30].

## A.5 Measuring and visualizing disentanglement

We measured disentanglement using the DCI metric of Eastwood and Williams [26]. DCI defines three different metrics: disentanglement, completeness and informativeness. For our purposes, we are only interested in the first, but they are all computed in a similar way. Disentanglement here is defined as the degree to which a latent variable represents at most one and only one generative factor.

Let $v_j$ be the ground truth values for each factor $g_j$. Assuming a trained model, the metric works as follows:

1. Compute the latent representation $z_n = \text{encode}(x_n)$ for all images $x_n$ in the dataset.

2. For each factor $g_j$ solve a regression problem with the $z_n$'s as covariates.

3. Construct a matrix $C$, where each index, $c_{ij}$, is the absolute value of the regression coefficient between the latent variable $i$ and generative factor $j$.

4. $C$ is then used to quantify the deviation of the latent representation from the ideal one.

The authors propose using Lasso regression or Random Forests (they provide a method to determine the coefficient for the latter). In this work we use Lasso with their proposed hyper-parameters. To compute the disentanglement score, the coefficients in $C$ are treated as a distributions, one for each column (latent), from which the entropy is computed:

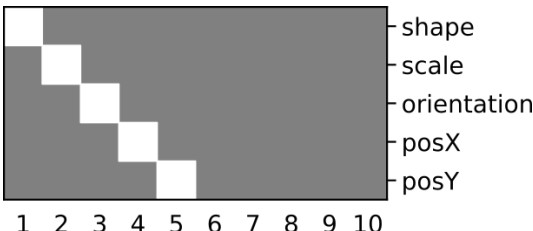

Figure 8: **Ideal disentanglement** Example of ideal disentanglement computed with the DCI metric for the dSprites dataset.

$$H(P_i) = -\sum_{j=0}^{J} P_{ij} \log_J P_{ij}$$

where $P_{ij} = C_{ij} / \sum_{k=0}^{J} C_{ik}$, where $J$ denotes the number of rows in $C$. The score for one variable is computed as:

$$d_i = 1 - H(P_i)$$

The $d_i$'s are then be averaged to obtain the final, overall disentanglement score:

$$D = \sum_i \rho_i \cdot d_i$$

$$\rho_i = \frac{\sum_{j=0}^{J} C_{i,j}}{\sum_{l,j} C_{l,j}}$$

where the $\rho_i$ are computed so as to account for dead units in the representation. We refer to $D$ as the disentanglement score throughout the main text, in line with most research on the topic.

DCI is interesting for a couple of reasons. First, it computes it's scores based on the ability to predict values from the latent representations. Additionally, the coefficients $C$ can be visualized using Hinton matrices, which means we can use it for both quantitative measurements and for visual inspection of the trained models, as we see below. An ideal model in this framework will produce a matrix $C$ that is diagonal, up to some permutation of the latent variables and disregarding any extra dimensions used in the latent representation (see Figure 8 for an example). This fits well with intuitions regarding disentanglement and with the definition proposed in Higgins et al. [29]. Given these coefficients, it is easy to assign latents to the corresponding generative factor. To avoid assigning a latent to two factors, we used the Munkres algorithm [31] as in Hyvarinen and Morioka [20].

For a highly disentangled model we can use this assignment to visualize pairs of latent variables in a 2D plot. Given to factors of interest (say 'shape' and 'position') we the latent variables that were assigned to them by the above procedure for each point in the dataset. This gives us a matrix of size $Nx2$ we can then compute a kernel-density estimate of the distribution of these points. Highly disentangled models should look like a regular lattice of almost Gaussian distributions. Models that are not as disentangled can have different properties, such as drifting or increased variance of the estimated kernel densities.

### A.6 Implementation

All the models and tasks where implemented in PyTorch (version $\geq 3.8$[3], Paszke et al. [32]) and Numpy [33]. The experiments where performed with the aid of the Ignite and Sacred frameworks [34, 35]. Visualizations were created using Matplotlib [36] and Seaborn [37]. Code required to reproduce the results can be found at `https://github.com/mmrl/lost-in-latent-space`.

We trained a total of $48 + 9$ models accounting for all combinations of datasets (4, dSprites, 3DShapes, MPI3D and Circles), losses (2, VAE and WAE), conditions (3, baseline with all the data, one success and one failure) and decoder architectures (standard or Spatial Broadcast) for the semi-supervised

---

[3]Note that 3.8 is a hard requirement here if we want to access the `tile` function for the Spatial Broadcast Decoder. Otherwise it must be implemented in Numpy and converted to a PyTorch tensor manually.

case, and the same combination of datasets and conditions for the latent prediction one (excluding the circles dataset). Each model took between 2 and 5 hours to train depending on the condition, except the latent-prediction case and the the circles dataset (under 1 hour). This totals to roughly 150 hours of wall clock time for training. Both this and the analysis were performed on a single workstation using an Intel Core i7-9700K CPU, with 32 GiB of RAM and an NVIDIA RTX 2080Ti GPU.

# B  Results

Below we provide Model scores (that were used to measure success at reconstruction as well as degree of disentanglement), some example reconstructions for each type of model and Hinton matrices that are useful to visualising the degree of disentanglement. We also provide visualisations of the latent space (akin to the ones presented in the main text). In each case, we present examples of two types of training objective (VAE & WAE – see Section A.4) and two types of interpolation (fixed and learned – see Equations 5 and 4) – making a total of four models for each condition (except for the MPI3D dataset where only models trained with the WAE objective succeeded in learning the task).

## B.1  Model scores

Table 2: **Scores for dSprites failure condition** The scores for the condition `[shape=square,posX>0.5]`. Reconstruction loss for VAEs are computed with pixel-wise binary cross-entropy (Bernoulli loss). For the WAE it is the pixel-wise mean squared error.

| Model | Train loss | Test loss | Disentanglement |
|---|---|---|---|
| VAE + learned interp | 8.353618 | 183.369238 | 0.881088 |
| VAE + fixed interp | 7.524311 | 366.997949 | **0.999999** |
| WAE + learned interp | 2.310111 | 44.854529 | 0.917322 |
| WAE + fixed interp | 2.038967 | 45.108577 | **0.983566** |

Table 3: **Scores for 3DShapes failure condition** The scores for the condition `[shape=pill,object hue={blue, purple}]`. Reconstruction loss for VAEs are computed with pixel-wise binary cross-entropy (Bernoulli loss). For the WAE it is the pixel-wise mean squared error.

| Model | Train loss | Test loss | Disentanglement |
|---|---|---|---|
| VAE + learned interp | 3459.323259 | 5050.596667 | **0.980030** |
| VAE + fixed interp | 3460.841481 | 5517.578000 | **0.960200** |
| WAE + learned interp | 5.905513 | 214.398937 | **0.999999** |
| WAE + fixed interp | 7.470970 | 224.247479 | **0.973512** |

Table 4: **Scores for 3DShapes success condition** The scores for the condition `[wall hue={red, orange},floor hue{purple, magenta}]`. Reconstruction loss for VAEs are computed with pixel-wise binary cross-entropy (Bernoulli loss). For the WAE it is the pixel-wise mean squared error.

| Model | Train loss | Test loss | Disentanglement |
|---|---|---|---|
| VAE + learned interp | 3467.451818 | 3498.955556 | **0.975767** |
| VAE + fixed interp | 3471.045758 | 3510.643056 | 0.893841 |
| WAE + learned interp | 5.079037 | 6.359890 | **0.972172** |
| WAE + fixed interp | 5.932047 | 32.906424 | 0.948327 |

Table 5: **Scores for MPI3D failure condition** The scores for the condition `[object shape=cylinder,vertical axis >0.5]`. Reconstruction loss for VAEs are computed with pixel-wise binary cross-entropy (Bernoulli loss). For the WAE it is the pixel-wise mean squared error.

| Model | Train loss | Test loss | Disentanglement |
|---|---|---|---|
| WAE + learned interp | 2.114849 | 4.598420 | 0.675261 |
| WAE + fixed interp | 1.586387 | 5.227437 | **0.976362** |

Table 6: **Scores for MPI3D failure condition** The scores for the condition `[shape={cylinder,sphere}, background color=salmon]`. Reconstruction loss for VAEs are computed with pixel-wise binary cross-entropy (Bernoulli loss). For the WAE it is the pixel-wise mean squared error.

| Model | Train loss | Test loss | Disentanglement |
|---|---|---|---|
| WAE + learned interp | 2.014464 | 2.271657 | 0.702668 |
| WAE + fixed interp | 1.694517 | 2.517065 | 0.966975 |

Table 7: **Scores for the Circles dataset** The scores for the two conditions in the score dataset. The "corner" condition excludes instances with `[posX>0.5, posY>0.75]`. The condition "midpos" excludes those with `[0.35<posX<0.65,0.35<posY<0.65]`.

| Model | Condition | Train loss | Test loss | Disentanglement |
|---|---|---|---|---|
| VAE + fixed interp | corner | 113.977726 | 4313.172778 | **0.843828** |
| VAE + fixed interp | midpos | 115.894145 | 157.793549 | 0.781331 |

Table 8: **Scores for the Simple dataset** The scores for the two conditions in the score dataset. The "corner" condition excludes instances with `[posX>0.5, posY>0.75,shape=triangle]`. The condition "midpos" excludes those with `[0.35<posX<0.65,0.35<posY<0.65,shape=triangle]`.

| Model | Condition | Train loss | Test loss | Disentanglement |
|---|---|---|---|---|
| VAE + fixed interp | corner | 129.493923 | 523.572257 | 0.901404 |
| VAE + fixed interp | midpos | 129.802943 | 267.656094 | **0.936586** |

## B.2 Reconstructions

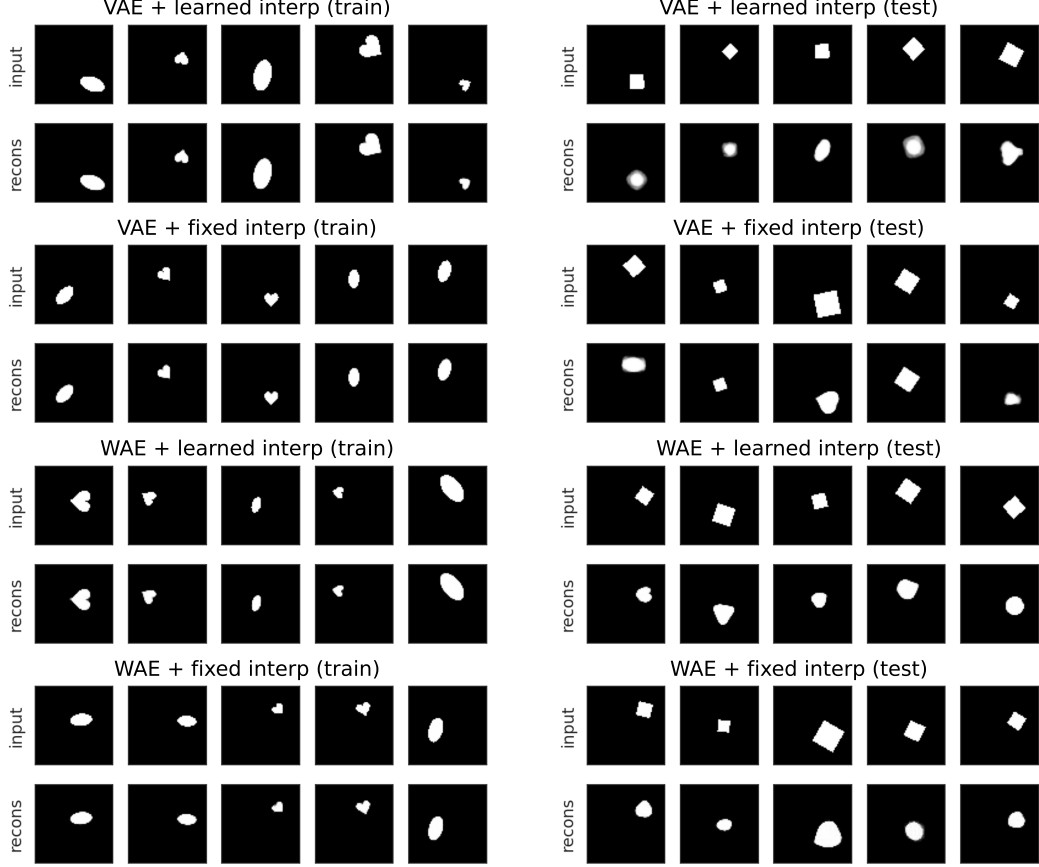

Figure 9: **Failures of combinatorial generalisation on `dSprites`** Some example reconstructions for the `dSprites` dataset for four models: (i) a VAE using learned interpolation (see Equations 4), (ii) a VAE using a fixed interpolation (see Equation 5), (iii) a WAE using a learned interpolation, and (iv) a WAE using a fixed interpolation. In each case, the five figures on the left show input (first row) and reconstruction (second row) for five training images. The five figures on the right show input and reconstructions for test images. These test images presented a novel combination of generative factors, here `[shape=square, posX` $> 0.5$`]` – that is, the model has seen all shapes on both the left and right hand side of the canvas, except squares, which have only been seen on the left hand side (posX $< 0.5$). All models learn representations that are highly disentangled. While reconstructions on the training dataset are really accurate - it is clear to see that the models struggle to reconstruct the test images, where a square is presented on the right hand side of the canvas.

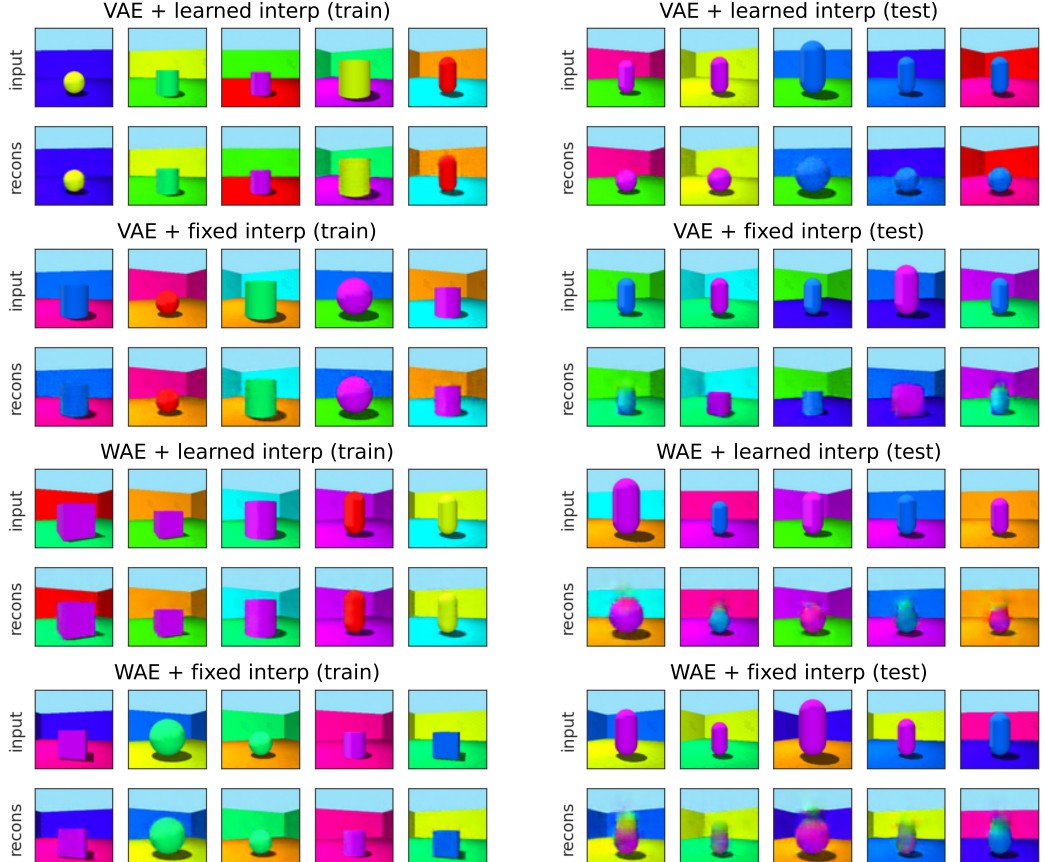

Figure 10: **Failures of combinatorial generalisation on** `3DShapes` Some example reconstructions for the `3DShapes` dataset for four models: (i) a VAE using learned interpolation (see Equations 4), (ii) a VAE using a fixed interpolation (see Equation 5), (iii) a WAE using a learned interpolation, and (iv) a WAE using a fixed interpolation. All models learn representations that are highly disentangled. In each case, the five figures on the left show input (first row) and reconstruction (second row) for five training images. The five figures on the right show input and reconstructions for test images. These test images presented a novel combination of generative factors, here `[shape=pill, object hue > 0.5]` – that is, the model has seen all the combinations of shapes and hues for the object in the middle of the image, except for the 'pill' shape, which has only been seen in "warmer" colors (object hue < 0.5). We observed that while reconstructions on the training dataset were really accurate - it is clear to see that the models struggled to reconstruct the test images, where the object was a 'pill' shape with a "cooler" color.

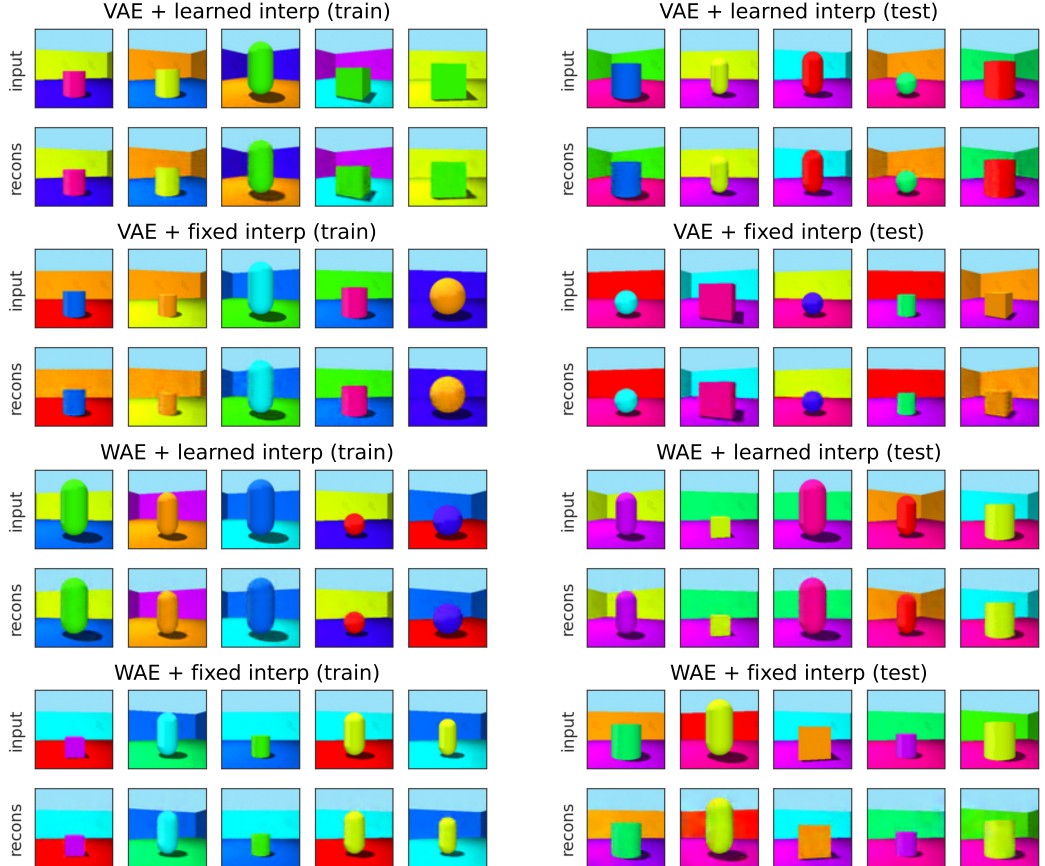

Figure 11: **Success in combinatorial generalisation on** `3DShapes` Some example reconstructions for the `3DShapes` dataset for four models: (i) a VAE using learned interpolation (see Equations 4), (ii) a VAE using a fixed interpolation (see Equation 5), (iii) a WAE using a learned interpolation, and (iv) a WAE using a fixed interpolation. All models learn representations that are highly disentangled. In each case, the five figures on the left show input (first row) and reconstruction (second row) for five training images. The five figures on the right show input and reconstructions for test images. These test images presented a novel combination of generative factors, here `[floor hue` $< 0.25$`, wall hue` $> 0.75$`]` – that is, the model has seen all wall hues and floor hues in the range $[0, 1]$, but it has never seen a combination a floor with a hue $< 0.25$ with a wall of a hue $> 0.75$. We observed that, in this case, the model succeeded at combinatorial generalisation, reconstructing the images equally well on the training and test sets (compare with Figure 10).

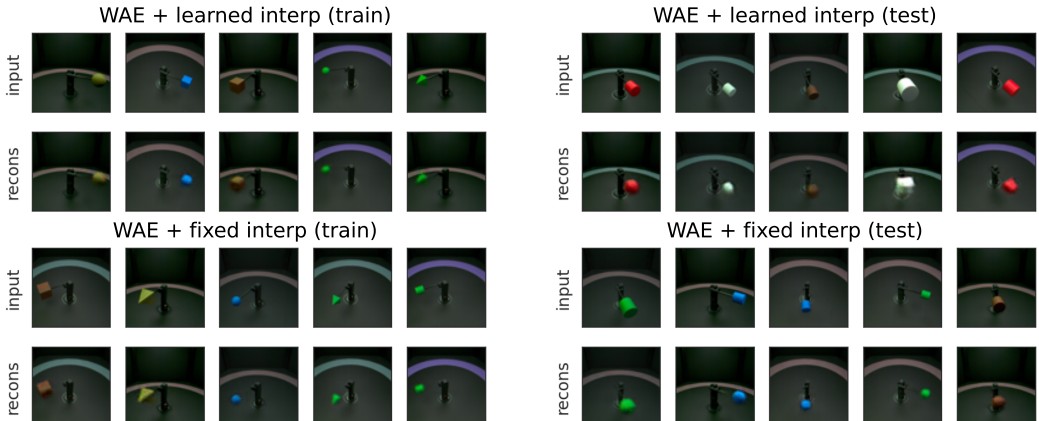

Figure 12: **Failures of combinatorial generalisation on** `MPI3D` Some example reconstructions for the `MPI3D` dataset for two models: (i) a WAE using learned interpolation (see Equations 4), (ii) a WAE using a fixed interpolation (see Equation 5). Both models learn representations that are highly disentangled. In each case, the five figures on the left show input (first row) and reconstruction (second row) for five training images. The five figures on the right show input and reconstructions for test images. These test images presented a novel combination of generative factors, here `[shape=cylinder, vertical axis > 0.5]` – that is, the model has seen all the combinations of shapes (of the object at the end of a rod) and positions on the vertical axis, except for the cylinder, which has only been seen at vertical axis positions $< 0.5$). We observed that while reconstructions on the training dataset were really accurate - both models struggled to reconstruct the test images with the left out combination of shape and position, frequently replacing the shape of the object.

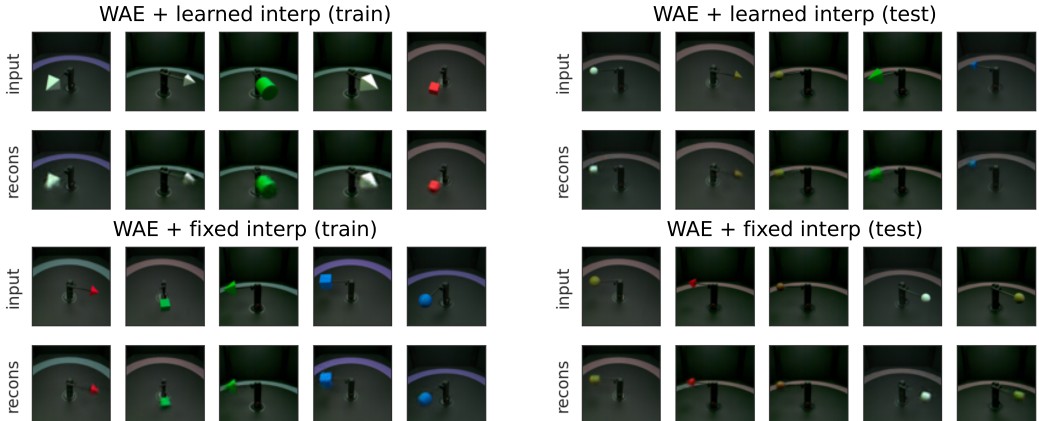

Figure 13: **Success in combinatorial generalisation on** `MPI3D` Some example reconstructions for the `MPI3D` dataset for two models: (i) a WAE using learned interpolation (see Equations 4), (ii) a WAE using a fixed interpolation (see Equation 5). Both models learn representations that are highly disentangled. In each case, the five figures on the left show input (first row) and reconstruction (second row) for five training images. The five figures on the right show input and reconstructions for test images. These test images presented a novel combination of generative factors, here `[shape={cylinder,sphere}, background color=salmon]`. We observed that, in this case, the model managed to successfully perform the task of combinatorial generalisation, reproducing the training as well as the test images with really good accuracy (compare with 12).

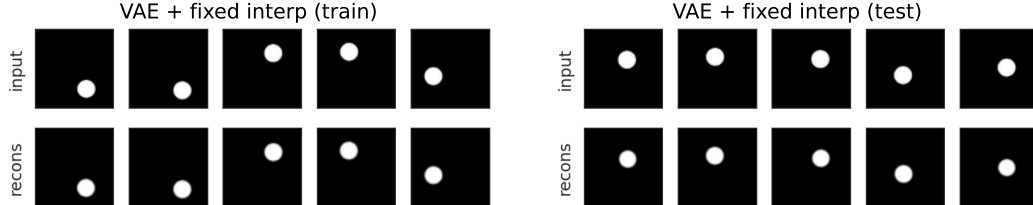

Figure 14: **Success in combinatorial generalisation on** `Circles` **dataset** Some example reconstructions for the `Circles` dataset. The model learns representations that are highly disentangled using fixed interpolation (see Equation 5). The five figures on the left show input (first row) and reconstruction (second row) for five training images. The five figures on the right show input and reconstructions for test images. These test images presented a novel combination of generative factors, here $[0.35 < \texttt{posX} < 0.65,\ 0.35 < \texttt{posY} < 0.65]$ – that is, the model has seen circles in all x-positions, as well as all y-positions, but never for a combination of x & y positions that fall in a central patch. We observed that the models managed to successfully perform the task of combinatorial generalisation, reproducing the circle in all positions of the canvas.

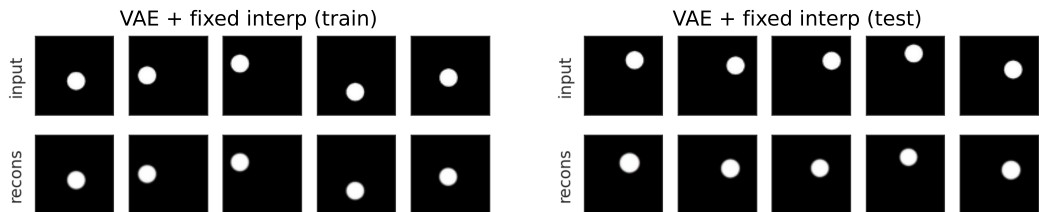

Figure 15: **Failure in combinatorial generalisation on** `Circles` **dataset** Some example reconstructions for the `Circles` dataset. The model learns representations that are highly disentangled using fixed interpolation (see Equation 5). The five figures on the left show input (first row) and reconstruction (second row) for five training images. The five figures on the right show input and reconstructions for test images. These test images presented a novel combination of generative factors, here $[\texttt{posX} > 0.5,\ \texttt{posY} > 0.5]$ – that is, the model has seen circles in all x-positions, as well as all y-positions, but never for a combination of x & y positions that fall in a top right corner. We observed that the models failed in this condition in line with results in Watters et al. [6]

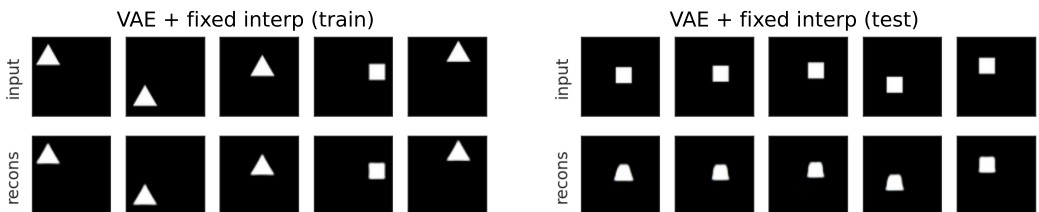

Figure 16: **Failures of combinatorial generalisation on** `Simple` **dataset for the middle patch condition** Some example reconstructions for the `Simple` dataset for two models: The model learns representations that are highly disentangled using fixed interpolation (see Equation 5). The five figures on the left show input (first row) and reconstruction (second row) for five training images. The five figures on the right show input and reconstructions for test images. These test images presented a novel combination of generative factors, here $[0.35 < \texttt{posX} < 0.65,\ 0.35 < \texttt{posY} < 0.65, \texttt{shape=triangle}]$ – that is, the model has seen triangles in all x-positions, as well as all y-positions, but never for a combination of x & y positions that fall in a central patch. It has seen squares in a central location. We observed that the models failed to properly generalise in this condition, as opposed to the corresponding condition in the circles dataset (see Figure 14)

## B.3 Disentanglement

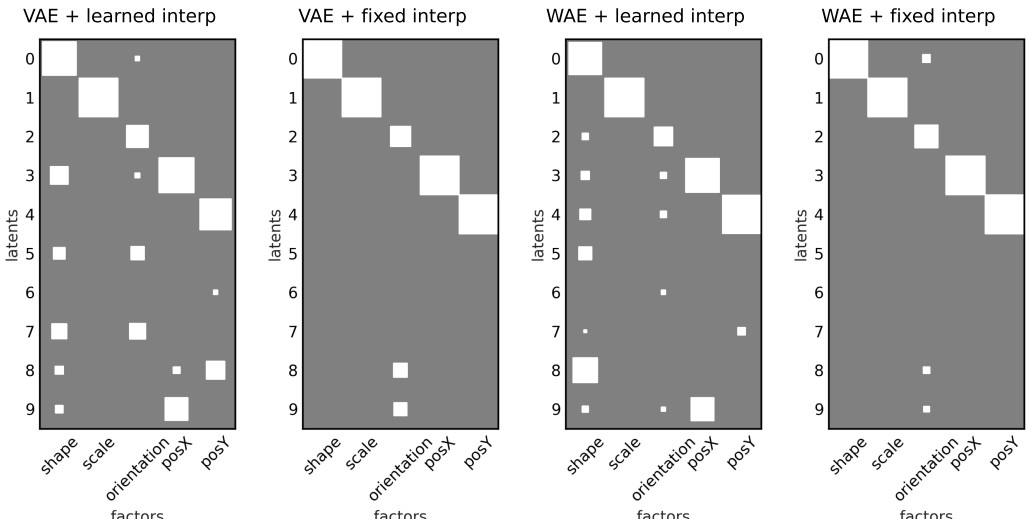

Figure 17: **Visualising disentanglement for** `dSprites` **dataset** Following Eastwood and Williams [26], we used Hinton matrices to visualise the extent of disentanglement. Each panel shows the matrix $C$ (see Section A.5) of regression coefficients between latent variables and generative factors. A disentangled model should have high coefficients for each pair of generative factors and latent variable but low coefficients everywhere else – i.e. the matrix should be sparse (see ideal Hinton matrix in Figure 8). Here we can see that all models learning the composition task show highly sparse matrices and models using fixed interpolation (Equation 5) achieve particularly high disentanglement.

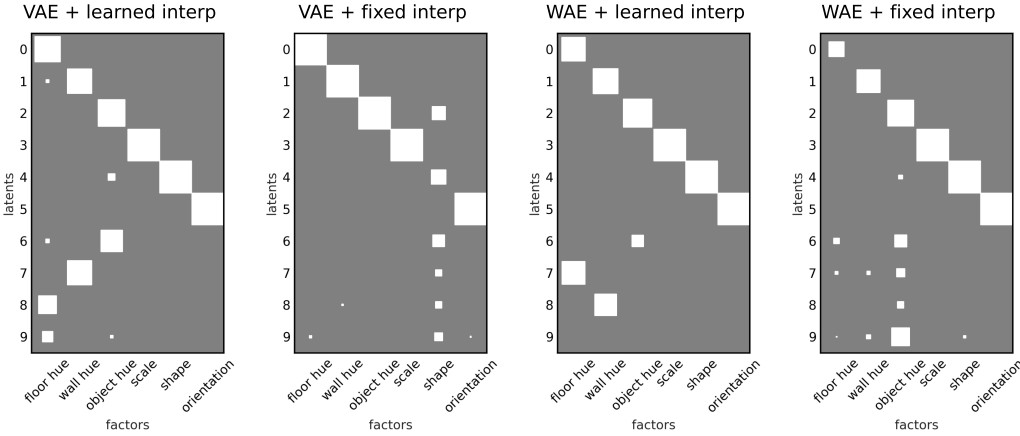

Figure 18: **Visualising disentanglement for** `3DShapes` **dataset, failure condition** Following Eastwood and Williams [26], we used Hinton matrices to visualise the extent of disentanglement. Each panel shows the matrix $C$ (see Section A.5) of regression coefficients between latent variables and generative factors.

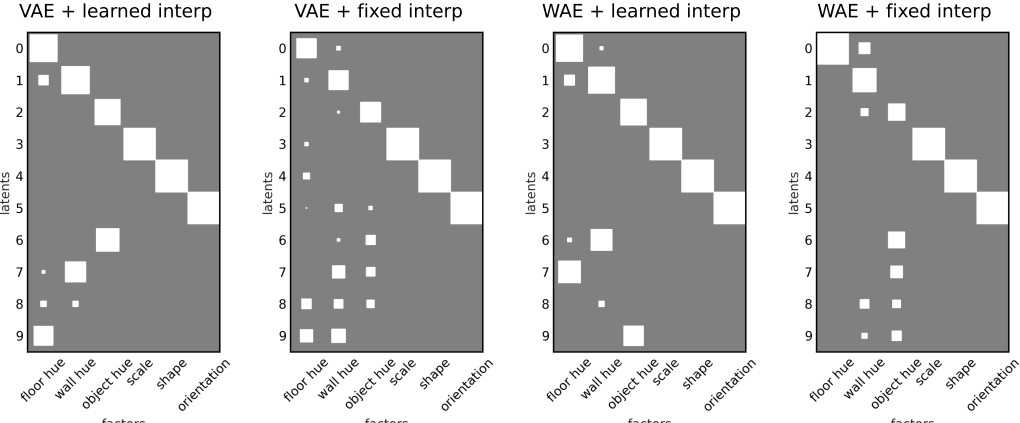

Figure 19: **Visualising disentanglement for** `3DShapes` **dataset, success condition** Following Eastwood and Williams [26], we used Hinton matrices to visualise the extent of disentanglement. Each panel shows the matrix $C$ (see Section A.5) of regression coefficients between latent variables and generative factors.

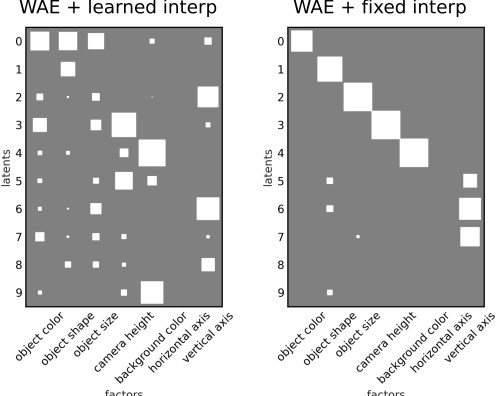

Figure 20: **Visualising disentanglement for** `MPI3D` **dataset, failure condition** Following Eastwood and Williams [26], we used Hinton matrices to visualise the extent of disentanglement. Each panel shows the matrix $C$ (see Section A.5) of regression coefficients between latent variables and generative factors.

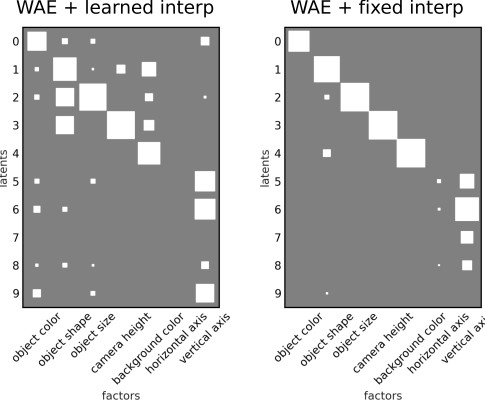

Figure 21: **Visualising disentanglement for** `MPI3D` **dataset, success condition** Following Eastwood and Williams [26], we used Hinton matrices to visualise the extent of disentanglement. Each panel shows the matrix $C$ (see Section A.5) of regression coefficients between latent variables and generative factors.

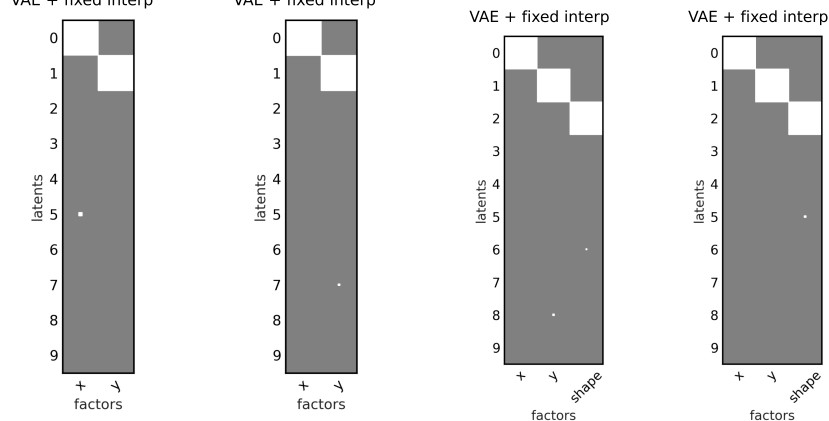

(a) `Circles` **dataset** Following Eastwood and Williams [26], we used Hinton matrices to visualise the extent of disentanglement. Each panel shows the matrix $C$ (see Section A.5) of regression coefficients between latent variables and generative factors. *Left* the corner condition. *The middle patch condition.*

(b) `Simple` **dataset** Following Eastwood and Williams [26], we used Hinton matrices to visualise the extent of disentanglement. Each panel shows the matrix $C$ (see Section A.5) of regression coefficients between latent variables and generative factors. *Left* the corner condition. *The middle patch condition.*

Figure 22: **Visualising disentanglement for the** `Circles` **and** `Simple` **datasets**

## B.4    Latent visualizations

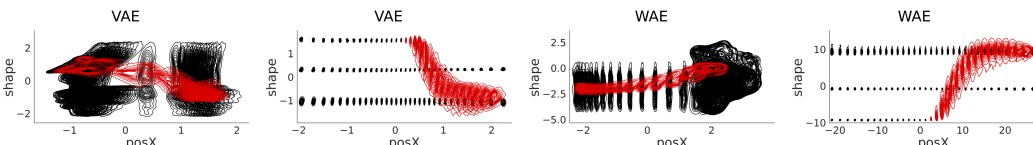

Figure 23: **Visualisation of latent space for `dSprites` dataset**. Distribution of encoded values for different combinations of shape and posX for the dSprites dataset. For models that do not disentangle well (first and third from left to right), we observe a very unstructured distribution of both training (black) and (test). Models with high disentanglement (second and fourth) show a very structured representation where it is easy to discriminate between different shapes and positions in the training set, but which diverge in the test set.

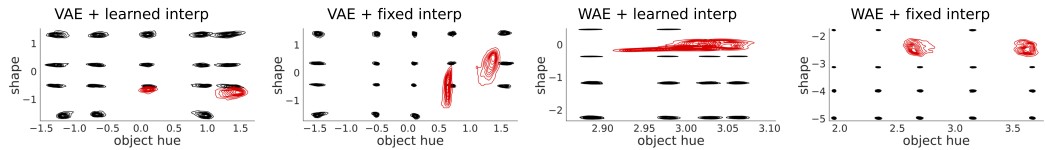

Figure 24: **Visualisation of latent space for `3DShapes` dataset, failure condition**. Distribution of encoded values for different combinations of shape and object hue for the 3DShapes dataset. In this case, all models showed high levels of disentanglement. The factor combinations for the training set are easily distinguishable. However, models show poor generalisation in the test set. The 4th model on the right is the one that shows the best results, yet the test distributions show a signfican increase in variance and drift towards observed training data.

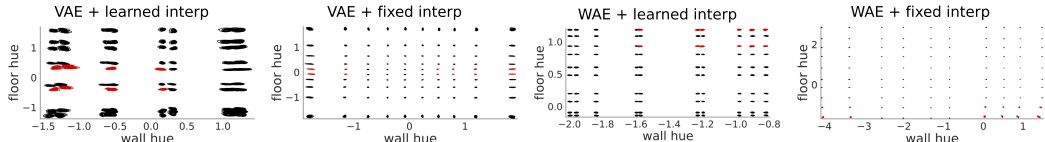

Figure 25: **Visualisation of latent space for `3DShapes` dataset, success condition**. Distribution of encoded values for different combinations of wall hue and floor hue for the 3DShapes dataset. In this case, all models showed high levels of disentanglement. The factor combinations for the training set are easily distinguishable. As opposed to the previous condition, models show excellent generalisation to the test data.

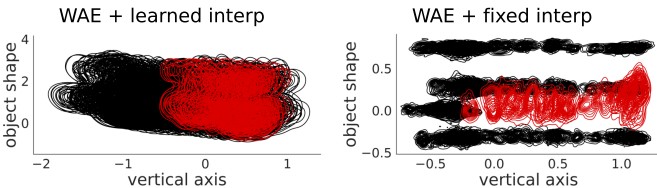

Figure 26: **Visualisation of latent space for `MPI3D` dataset, failure condition**. Distribution of encoded values for different combinations of shape and vertical axis for the MPI3D dataset. The model on the left shows poor disentanglement with the joint distribution of the best match latents for both training (black) and testing examples (red) showing significant overlap. The model on the right on the other hand achieved high disentanglement. The encoded value for each shape is easily distinguishable. The encoded angle of the arm is not as clearly separated. The test data shows a significant drift, though not as dramatic as in `dSprites`.

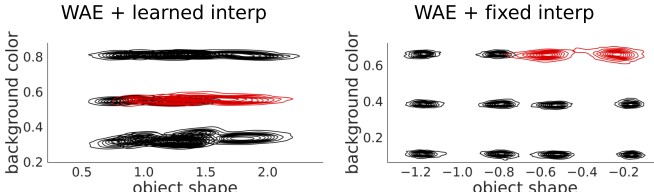

Figure 27: **Visualisation of latent space for** `MPI3D` **dataset, success condition**. Distribution of encoded values for different combinations of shape and vertical axis for the MPI3D dataset. The first three models from the left show poor disentanglement with the joint distribution for both training (black) and testing examples (red). The model on the right on the other hand achieved high disentanglement. The encoded value for each shape background color combination is easily distinguishable. There is no hint of drift or even increase in variance.

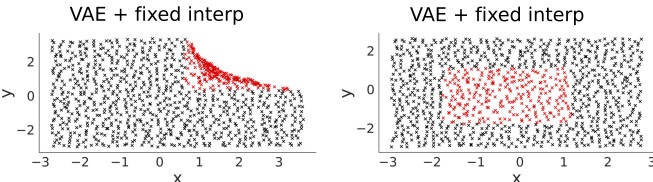

Figure 28: **Visualisation of latent space for** `Circles` **dataset**. We replicate the results from Watters et al.. On the left, the Spatial Broadcast Decoder shows poor generalisation when tested on the corner case. Test examples (red) bend over the closest seen training examples (black). On the right, when the excluded patch is locate in the center, the model shows excellent generalisation.

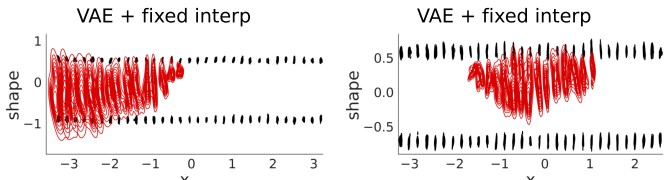

Figure 29: **Visualisation of latent space for** `Simple` **dataset**. We show that the Spatial Broadcast Decoder [6] fails when we introduce another shape in the middle patch position. On the left, the result for the corner case, showing again poor generalisation. The drift is similar to the one observed in dSprites: as we move away from observed positions the latent representation experience a dramatic increase in variance and shift towards the mean of the other shape. On the right, when the excluded patch is locate in the center, the model also fails as the introduction of a new shape makes turns the task too hard for the model.

# C   Interactive vs Non-interactive combinations of generative factors

In our simulations, we observed that models failed at combinatorial generalisation when the generative factor 'shape' was combined with another generative factor (position / color). We hypothesised that this was because shape combines with other factors (position / color / orientation) in an *interactive* manner – that is, the value of any pixel in the image is determined jointly by the value of `[shape]` and other factors. To understand why this is a hard problem, consider a graphical representation that captures the dependencies between the pixels of an image and generative factors (Figure 6.a). Each pixel and each generative factor corresponds to a node in this graph – pixel nodes are at the bottom and generative factor nodes are at the top. The value of the pixel nodes depends on the value of one or more generative factors. We capture this dependence as edges between pixel nodes and generative factor nodes.

Consider two different conditions. In the first condition, we take a combination of the generative factors `[floor-hue, wall-hue]`. In this case, the factor node `[floor-hue]` will determine one set of pixel nodes, while the factor `[wall-hue]` will determine another (mutually-exclusive) set of pixel nodes. During training, the model needs to learn how to map the value of each generative factor to the corresponding pixel. Once it has learned this mapping, it can easily generalise to unseen combinations of `[floor-hue, wall-hue]` because the same graphical dependencies learned from training work. We call this case the *non-interactive* condition.

Now, consider a second condition where the generative factors are `[shape, posX]`. In this case, the value of each pixel node is *jointly* determined by the values of both generative factors. Furthermore, note that the mapping is highly nonlinear and complex. For example, in the dSprites dataset, a pixel node may have value $+1$ for a given value of `[shape]` and `[posX]`, but this value may change to $-1$ for a slight change in either factor. Other changes will have no effect on the value of some pixels. To succeed at combinatorial generalisation, a model must account for these dependencies and any changes that occur dynamically as a result of the interaction between the generative factors. We call this condition *interactive* and hypothesised that models will succeed in the non-interactive condition, but succeed in the interactive condition. The results in Figures 3 and 7 confirm this hypothesis.

# D    Control experiments

## D.1    Rebalanced datsets

In all experiments where we tested combinatorial generalisation, combinations along a subset of dimensions were excluded from the training set. This means that the model was trained on an unequal number of samples from different dimensions. Here we include results for simulations where the datasets have been rebalanced so that all shapes are seen the same number of times. Figures show that models achieve high levels of disentanglement while still failing to generalise. Since we observed that models achieve disentanglement more consistently compared to an unbalanced dataset, we rebalance the dataset in all subsequent experiments.

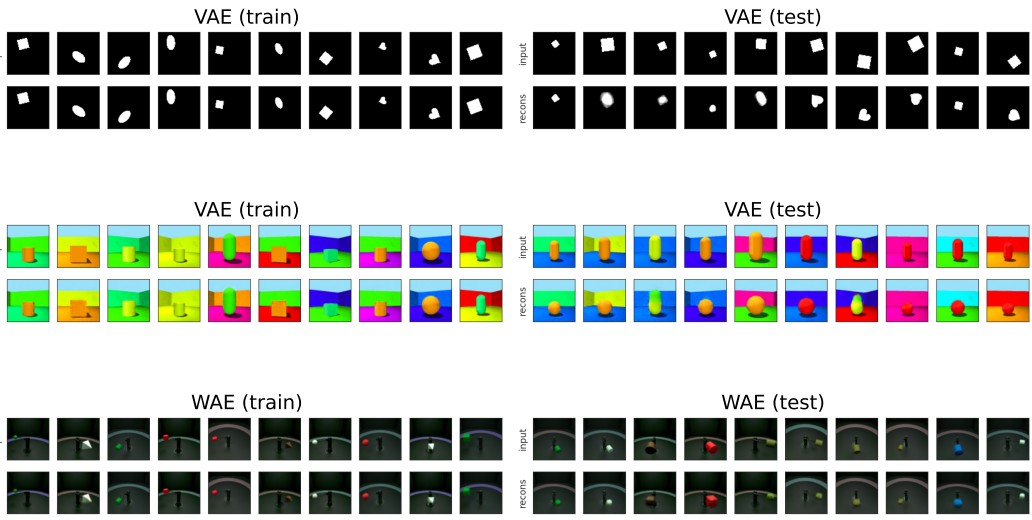

Figure 30: **Failures of combinatorial generalisation for models trained on dSprites, 3DShapes and MPI3d**. In each case, the top row shows input images and the bottom row shows reconstructions. The images on the left are sampled from the training set and the ones on the right from the test set. Failures of generalisation usually manifest as the model swapping the input shape. Only harder conditions with rebalanced datsets (i.e. all shapes are observed the same number of times) are shown.

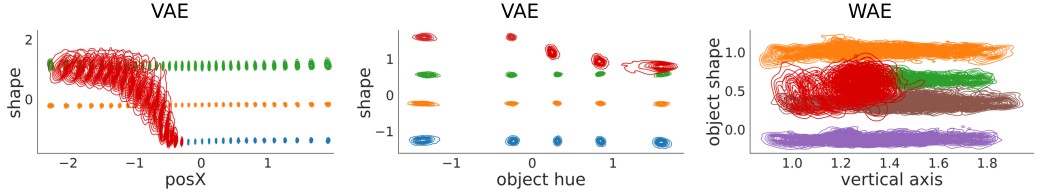

Figure 31: **Visualisation of latent space for models trained on dSprites, 3DShapes and MPI3d**. Like in other results, the latent space projections for test images are shown in red. We have also colour-coded the training set images according to shape. This makes it clear that the models are clearly disentangled (one shape (colour) maps to one row (value of latent variable)). Despite this, we see generalisation failures, where left-out combinations show an increase in variance and the mean drifts to values seen during the training set. Only harder conditions with rebalanced datsets (i.e. all shapes are observed the same number of times) are shown.

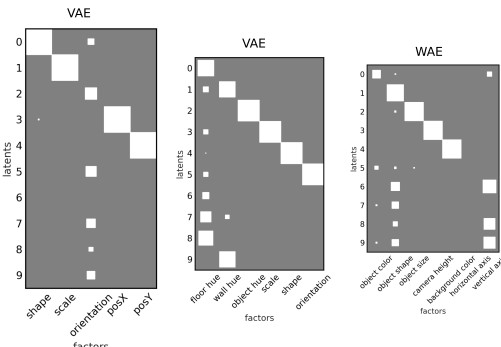

Figure 32: **Visualising disentanglement for models trained on dSprites, 3DShapes and MPI3d**. Only harder conditions with rebalanced datsets (i.e. all shapes are observed the same number of times) are shown.

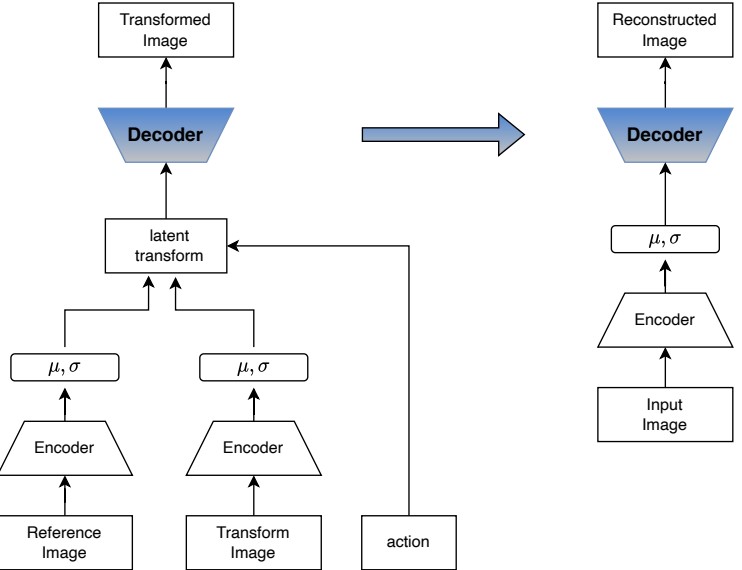

Figure 33: **Schematic illustration of the ideal decoder experiment.** The "ideal decoder" was constructed by training the model on the composition task (left) using the entire dataset (here dSprites) and then inserting the trained decoder from this setup into a VAE which was then trained on the image reconstruction task (right). All weights of the decoder were frozen, essentially limiting the learning to the encoder.

## D.2    Using an ideal decoder

For all results in the main text where we use the semi-supervised (composition) task, the model was trained end-to-end. This means that the encoder and decoder were jointly trained. Then at test time, we found that the model failed to reconstruct images containing left-out combinations of generative factors and the encoder also failed to project these images with left-out combinations to the correct region of the latent space. However, it could be argued that the failure to project images in the correct region of latent space is not entirely down to the encoder as the encoder and decoder were jointly trained. In other words, if one could fix the decoder – i.e. the decoder can learn to correctly map regions in the latent space to image space – then the encoder may learn to perform combinatorial generalisation. To test this hypothesis we ran the dSprites task using an "ideal decoder".

We constructed this ideal decoder (see Figure 33) by training a model in the usual way on the composition task, except the decoder was trained on the entire dataset (with no combinations left out). Once this model had been trained, we verified that the decoder had learned disentangled representations. Figure 34a visualises the latent space of this model showing that the model did indeed learn highly disentangled representations. We then took a new (untrained) VAE and replaced the decoder with this "ideal decoder" and froze all the weights. In the next step we trained the VAE on the image reconstruction task on the dSprites dataset. Critically, this time we left out some combinations of shape and position (the same combination that was left out in Figure 3). We then tested this model in the regular manner on samples from all possible combinations of shapes and positions. The results of the reconstructions are shown in Figure 34c. We can see that, just like the model where the encoder and decoder were jointly trained, the model manages to reconstruct the training images but fails in most cases to produce the test images accurately. Furthermore, a visualisation of the latent space (Figure 34b shows that, despite being trained with this ideal decoder, the encoder fails to map unseen combinations to the correct region of the latent space. These results verify that the failure of the encoder to project images with novel combinations is not the result of the decoder used to train the model.

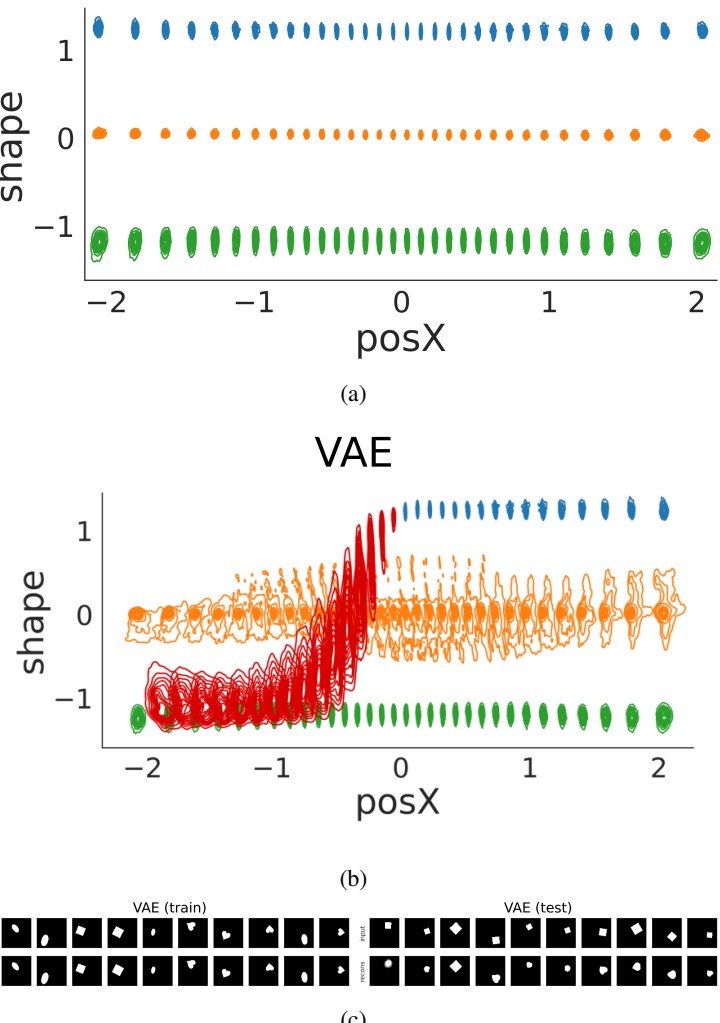

(a)

(b)

(c)

Figure 34: **Encoder trained with a perfect decoder**. We take a model that has achieved high level of disentanglement on the full dataset (top row) and freeze it's docoder. A new encoder is then attached to this perfect decoder and trained on the generalisation case for dSprites producing a set of latent representations (middle) and reconstructions for both training and test data (bottom row).

### D.3 Testing discrete latent representations

The CascadeVAE [27] model uses discrete latents which are supposed to learn to represent categories such as shape.

#### D.3.1 Encoder and Decoder architectures

All models used in the main text use continuous latent variables, which are the standard variables used for VAEs. However, our data contains various ordinal variables. For example, `[shape]` has a discrete set of values for the dSprites dataset. Some recent research shows that using a set of discrete variables leads to lower reconstruction loss and better disentanglement. To test whether using discrete latent variables also leads to better combinatorial generalisation for shape, we tested the CascadeVAE proposed by Jeong and Song [27]. This model uses discrete latent variables alongside continuous variables. To infer the values of the continuous ones, they use the standard approach in VAEs, producing mean and standard deviation values of a Gaussian distribution. For the discrete variables they use a top down iterative procedure which is defined by a nested optimization operation:

$$\max_{\theta,\phi} \left( \max_{d^{(1)},...,d^{(n)}} \sum_{i=1}^{n} \mu^{(i)^T} d^{(1)} - \lambda \sum_{i\neq j} d^{(i)^T} d^{(j)} \right)$$

$$- \beta \sum_{i=1}^{n} D_{KL}(q_\phi(z|x^{(i)}||p(z))$$

$$\text{subject to } ||d^{(i)}|| = 1, d^{(i)} \in 0, 1^S, \forall i,$$

where $\mu^{(i)}$ donotes the vector of the likelihood $\log p_\phi(x^{(i)}|z^{(i)}, e_k)$ evaluated at each discrete value $k \in [S]$.

The results for the dSprites dataset using CascadeVAE are shown in Figure 35. As can be seen from this figure, the model trains successfully for this dataset, showing impressive reconstruction. However, just like the models tested in the main text, it fails at combinatorial generalisation.

### D.4 Testing interaction between factors

Given our conclusion that failures to model certain interactions appropriately could be the cause of the failures of previous approaches, we tested a recent model – Commutative LieGroupVAE [28] – that uses an adaptive equivariant structure, rather than a fixed vector space, to learn factors of variation in the data. This approach combines explicit modeling group operations plus penalties to the learned basis in order to learn a highly disentangled representation of the input (see the original work for more details). The hope is that because this method is adaptive, it may be able to capture not only the generative factors underlying the data, but also the dependencies between them (see Figure 6).

To check whether this approach helps overcome the problem of combinatorial generalisation for interactive generative factors, we replicated our experiments with 3DShapes under the interactive and non-interactive conditions. The first simulation tested the condition where the combination `[floor hue` $< 0.25$`, wall hue` $> 0.75$`]` was left out of the training set. This is the non-interactive condition, since the `[floor-hue]` and `[wall-hue]` determine different parts of the image. The second simulation tested the condition where the combination `[shape=pill, object-hue`$> 0.5$`]` was left out of the training set. This is the interactive condition since `[shape]` and `[object-hue]` determine the same set of pixels in the image. In addition, we ran a third simulation, on the dSprites dataset, where the combination `[shape=square, posX>0.5]` was left out of the training set. This is also an interactive condition because shape and position jointly determine the pixels of the image.

Visualizing the latent space of this model is harder, since each factor is now represented in a subspace that is not one-dimensional, but ten-dimensional (we used the original code and parameters from the author's original repository). To plot latent representations, we first projected each subspace to a one-dimensional representation using PCA, and computed disentanglment scores on the result. Disentanglement scores for the 3DShapes simulations reached 0.9 on the interactive case. For each case we ran 5 seeds and have shown the best models in terms of disentanglement (they all achieved similar reconstruction scores).

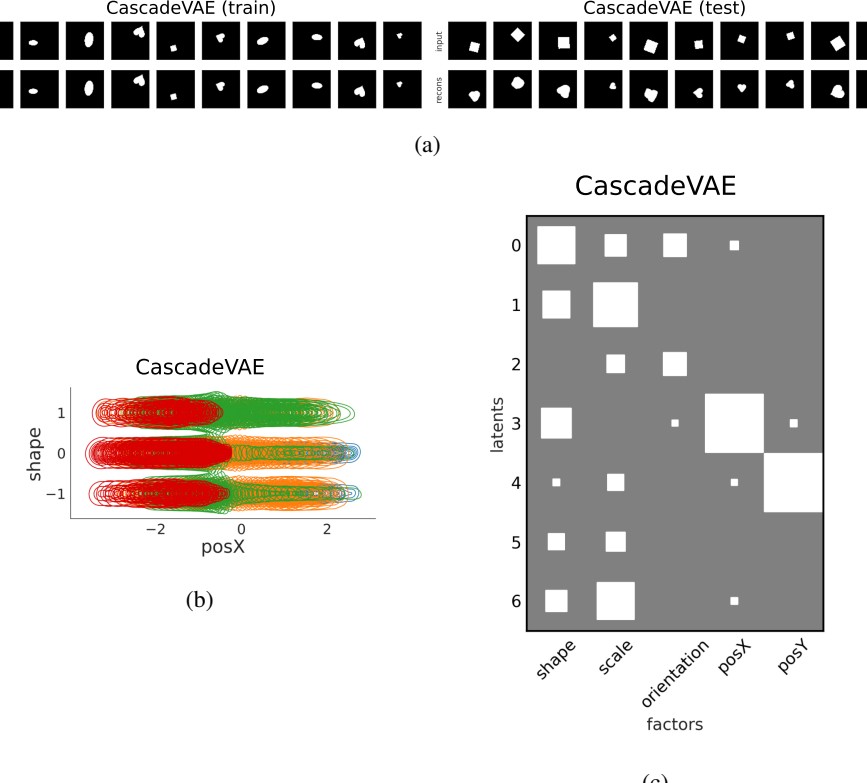

(a)

(b)

(c)

Figure 35: **Results for CascadeVAE**. (a) shows image reconstructions for training (left) and test (right) for an example model seed. As we can see from these reconstructions, the model successfully trains, reproducing the correct output image for each input image. However, it fails to reconstruct the unseen test combinations in a number of cases, replacing squares with hearts, etc. (b) show the latent space projection for training and test images. We have colour-coded each shape as a different colour and the test cases in red. We can again see that the model fails for the test cases, projecting the unseen shape (square) to three different values of the latent variable. However, it should also be noted here that, unlike the models discussed in the main text, this model does not show a high level of disentanglement, consequently mapping each shape to several values of the latent variable even for the training data. This lack of disentanglement is clear from examining the Hinton matrices in (c).

The results of these simulations are presented in Figures 36, 37 and 38. First of all, like the other models tested in the main manuscript, we observed that the LieGroupVAE manages to learn these tasks and performs image reconstruction to an impressive degree on the training set. Secondly, we found that this model succeeds on the non-interactive condition (`[floor hue < 0.25, wall hue > 0.75]`), reconstructing images with unseen combinations of floor-hue and wall-hue successfully (see Figure 36). Lastly, we found that, just like the other models discussed in the manuscript, LieGroupVAE also failed to reconstruct images in the interactive conditions – `[shape=pill, object-hue=> 0.5]` in 3DShapes (Figure 37) and `[shape=square, posX>0.5]` in dSprites (Figure 38). These simulations suggest that the approach used in LieGroupVAE is not sufficient to overcome the problem of combinatorial generalisation for interactive generative factors.

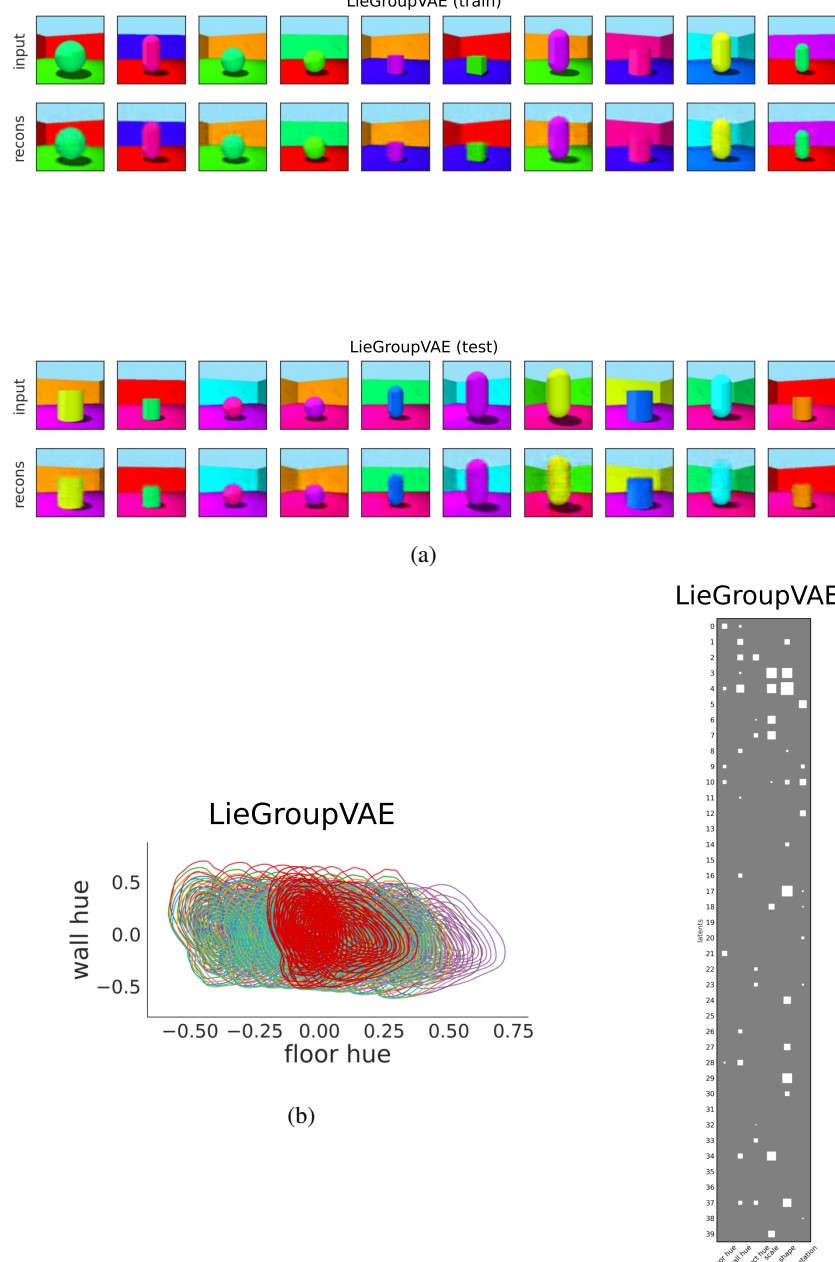

Figure 36: **Results for LieGroupVAE – non-interactive condition (3DShapes)**. (a) shows image reconstructions for training (top) and test (bottom) for an example model seed. As we can see from these reconstructions, the model successfully trains, reproducing the correct output image for each input image. It also succeeds at reconstructing the (unseen) test images in this non-interactive condition (note the combinations of wall-hue and floor-hue that are correctly reproduced). (b) show a visualisation of latent space. We have colour-coded each shape as a different colour and the test cases in red. It should also be noted here that, unlike the models discussed in the main text, the multidimensional nature of the subspaces for each subgroup makes it difficult to visualise these in a satisfactory way. (c) visualises the dependencies between generative factors and latent variables using Hinton matrices. In general, we obtained models that were highly disentangled, with disentanglement score 0.90. In this case the number of latents is much higher since the group size used in the original work is 400 for this dataset instead of 100 as used in dSprites.

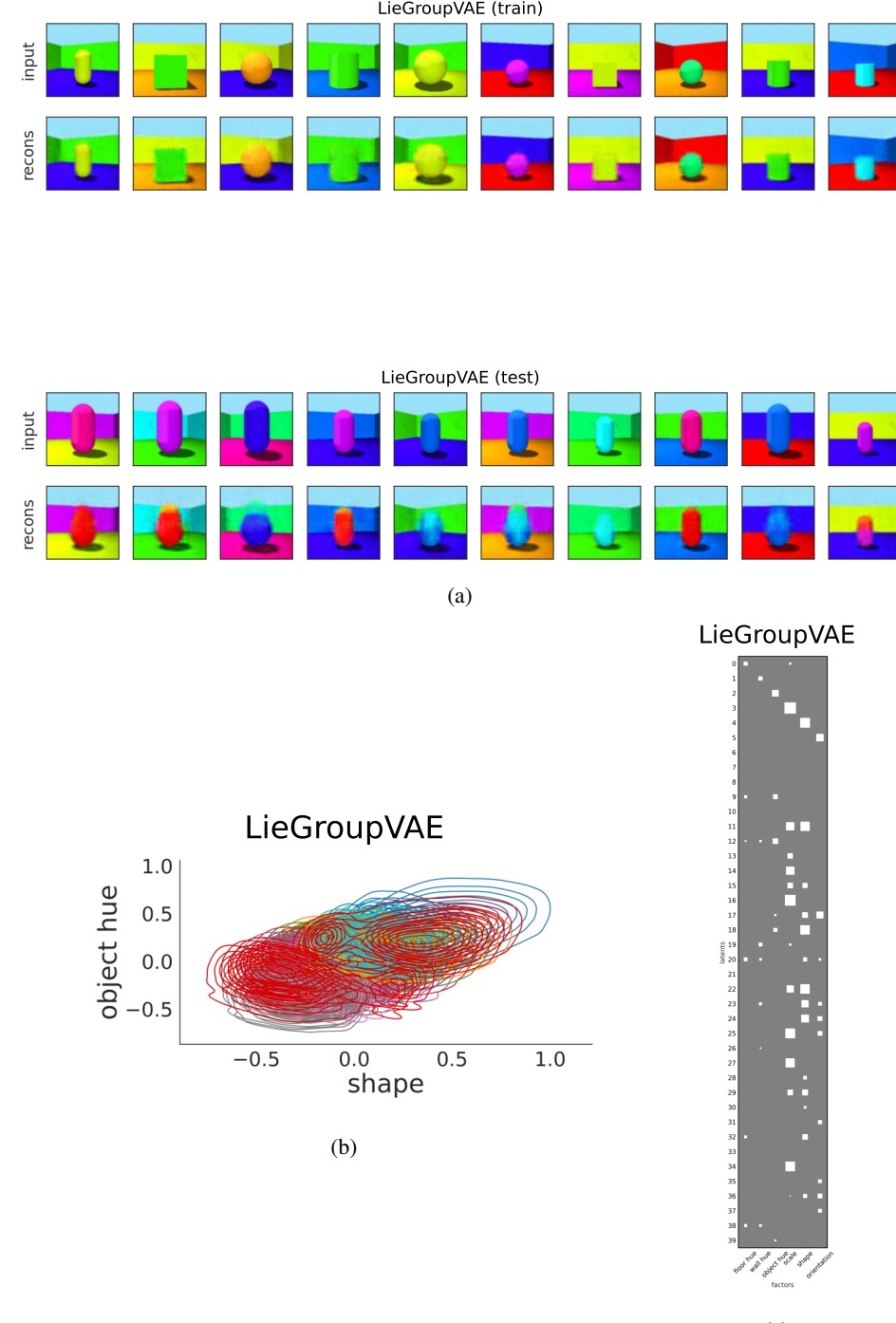

Figure 37: **Results for LieGroupVAE – interactive condition (3DShapes).** (a) shows image reconstructions for training (top) and test (bottom) for an example model seed. Again, the model successfully trains, reproducing the correct output image for each input image. However, in this case, it fails to reconstruct the unseen test combinations in a number of cases, though the mistakes are more varied than with other models, mixing shapes, colours and even combining them in some cases. (b) show the latent space projection for training and test images, where we have colour-coded each shape as a different colour and the test cases are in red. It should also be noted here that, unlike the models discussed in the main text, the multidimensional nature of the subspaces for each subgroup makes it difficult to visiaulize these in a satisfactory way.

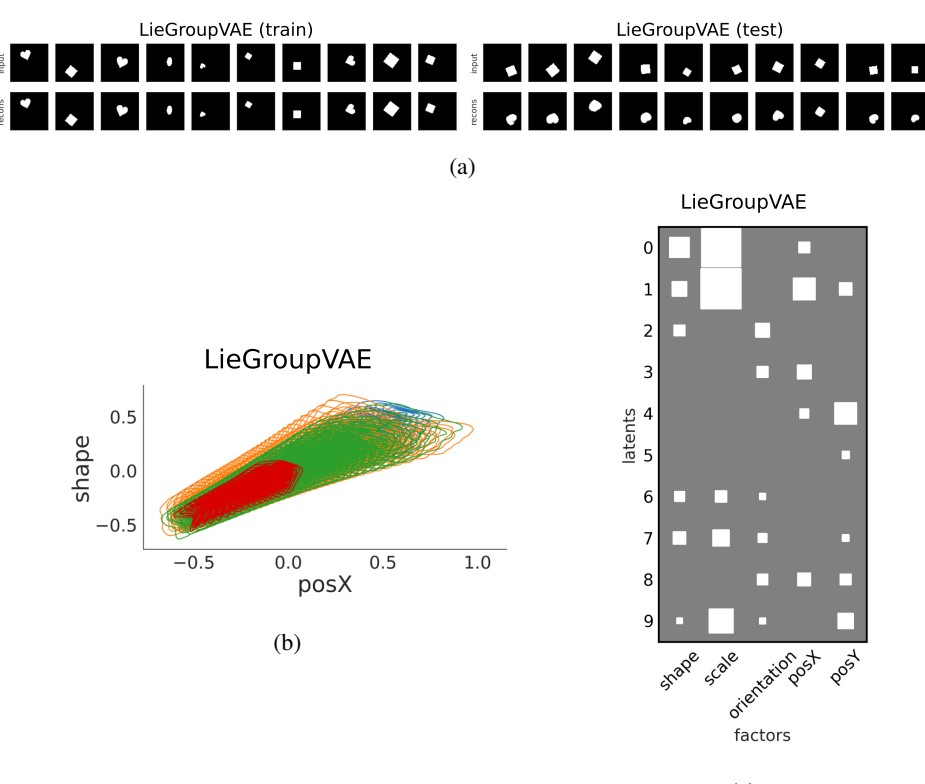

Figure 38: **Results for LieGroupVAE on dSprites**. (a) shows image reconstructions for training (top) and test (bottom) for an example model seed. Like in the case of 3DShapes (interactive condition), the model successfully trains, reproducing the correct output image for each input image. However, it fails to reconstruct the unseen test combinations in a number of cases, with squares being reconstructed as blobs that resemble hearts (b) show the latent space projection for training and test images. We have colour-coded each shape as a different colour and the test cases in red. (c) shows the Hinton matrices of these models. In all cases, models managed to achieve a disentanglement score $> 0.9$