# OpenReview forum: "Lost in Latent Space: Examining failures of disentangled models at combinatorial generalisation"
_NeurIPS.cc/2022/Conference — NeurIPS 2022 Accept_

### Official Review · Reviewer_HV8h · 2022-07-06

**Rating:** 6
**Confidence:** 5
**Soundness:** 4 excellent
**Presentation:** 4 excellent
**Contribution:** 2 fair

**Summary:**

The paper investigates the combinatorial generalization problem in disentangled representation learning. The paper confirms the generalization failures found by previous works, and investigated more detail about the reasons for these failures by considering whether they are due to encoding failure or decoding failure. From experiments, the paper shows that the factor ‘shape’ is the hardest for the models to generalize. This paper concluded with a hypothesis that these failures may be caused by the lack of a causal model that can model the interaction between factors.

**Questions:**

Some more investigation experiments on JointVAE, CascadeVAE, and LieGroupVAE would be great to address most of my questions mentioned in Weaknesses.

**Limitations:**

Yes

**Strengths And Weaknesses:**

Pros
1. The combinatorial generalization problem is a valuable problem to investigate.
2. The experiments presented in this paper are clear and well-conducted.
3. The visualization of the distribution between learned latent variables and generative factors (e.g. Fig 3 bottom) is beautiful and informative.

Cons
1. Only disentanglement models minimizing total correlation with continuous latent variables are considered. Other models should also be considered such as ControlVAE [1], CascadeVAE [2], and LieGroupVAE [3]. Particularly, as this paper shows that ‘shape’ is the most difficult factor to generalize while ‘shape’ is also a discrete factor, it is more valuable to investigate if models with discrete latent variables (e.g. JointVAE [4] and CascadeVAE [2]) can avoid this type of failure.
2. Paper is simple but somehow lacks novelty to me. The combinatorial generalization problem is known. This paper confirms this finding and shows the failures happen in encoding, but no much further insights can be drawn. This work could be more complete if it can show some potential evidence of solutions to this problem.
3. The paper concludes with a hypothesis that a model may be helpful if it can consider the interaction between factors. The LieGroupVAE [3] tried to model disentanglement from the perspective of matrix composition, which may be seen as an approximation of interaction between generative factors. It would be interesting if this paper can also investigate how this model performs on this combinatorial generalization problem.

[1] ControlVAE: Controllable Variational Autoencoder

[2] Learning Discrete and Continuous Factors of Data via Alternating Disentanglement

[3] Commutative Lie Group VAE for Disentanglement Learning

[4] Learning Disentangled Joint Continuous and Discrete Representations

---

> ### Author Response · Authors · 2022-08-02
> **Response to review**
>
> R4.1. Models with discrete factors: Thank you - this was a really helpful suggestion. Obviously, given the limited time we cannot test all the models mentioned by the reviewer. But we have tested a subset of our results for CascadeVAE. The reason why we chose CascadeVAE was that, (i) as the reviewer suggests, it uses discrete latent variables to code shape, (ii) it is a more recent model than JointVAE and inherits some of it's properties, and (iii) the authors claim it leads to large disentanglement of latent variables. We have included the results for this model on the dSprites task in Figures 35 in Appendix D and discussed them in greater detail in point G1 above.
>
> R4.2. Novelty: We agree that it would have been really great to propose a solution to the problem. However, for reasons listed in point G5 above, we don't think the solution is trivial. We also believe that identifying limitations of models through careful experimentation is a key part of the scientific process.
>
> R4.3. LieGroupVAE: Thank you for making us aware of this model, which we had not come across before. We did try to implement this. However, given to the short amount of time we have not been able to implement this particular model. If the reviewer is keen for us to try this model, we can  attempt to provide these results by the end of the discussion period.

---

> > ### Comment · Reviewer_HV8h · 2022-08-07
> > **Thank Authors for the Response**
> >
> > Thank you very much for the response. I will revise my rating accordingly.
> >
> > The CasecadeVAE experiment is an excellent addition to the paper. It addressed my concern about discrete latent variables in this disentanglement generalization problem.
> > It would be even better if the authors could also provide experiments on the LieGroupVAE so that some initial experimental discussions on the assumption of "interaction between factors" is touched upon.

---

> > > ### Author Response · Authors · 2022-08-08
> > > **LieGroupVAE experiments**
> > >
> > > Thank you for your feedback. We are really pleased to hear that you found the results for CascadeVAE informative.
> > >
> > > We have now finished simulating the results for LieGroupVAE and updated the manuscript accordingly (Section D4 in the Supplementary materials). We found that the results of this model replicate our previous findings – the model succeeded at the non-interactive combinatorial generalisation for 3DShapes, but failed at the interactive generalisation on both 3DShapes as well as dSprites. We would like to note here that the latent space of this model is more complex than other models we have simulated as each generative factor is projected to a subspace that is no longer one-dimensional. This makes visualisation of the latent space somewhat unsatisfactory. Nevertheless, we managed to train models with disentanglement scores in excess of 0.9 (i.e. highly disentangled) and these models showed the same pattern of results observed for other models in the manuscript.
> > >
> > > We would like to thank the reviewer again for their suggestions, and we hope that these set of simulations address their concerns.

---

> > > > ### Comment · Reviewer_HV8h · 2022-08-08
> > > > **Thanks for your response**
> > > >
> > > > Thanks for your response. The extra experiments are very informative and addressed my concerns. I am happy to raise my rating.

---

### Official Review · Reviewer_DoYc · 2022-07-10

**Rating:** 6
**Confidence:** 4
**Soundness:** 3 good
**Presentation:** 3 good
**Contribution:** 3 good

**Summary:**

The paper studies generalization failures of models with disentangled latent spaces to unseen combinations of factors. The authors present experiments aiming to determine whether the encoder or decoder, or other architectural aspects are causing these failures. To this end, the paper presents latent space visualizations and evaluations on the DCI disentanglement metric. Furthermore, the authors observe that the way the factors interact impacts whether they generalize in combination with other factors.

**Questions:**

I don’t have any specific questions, but I would appreciate comments on the points raised above that I believe could provide deeper insights into the generalization failures observed in the paper.

**Limitations:**

I feel that the paper could provide deeper insights into why the observed failure modes occur, please see the “strength and weakness” section above.


**Strengths And Weaknesses:**

The paper addresses an important problem in disentanglement learning, namely that of combinatorial generalization. The paper confirms and extends observations from prior work. I see the following as strong points of the paper:
- The authors consider different model/architecture variants (VAE, WAE, spatial broadcasting generator) on top of the composition task from [25] and observe similar generalization failures for all models.
- The observation that factors which “do not interact” with others (e.g. floor color and wall color in 3DShapes) better generalize to unseen combinations than factors that interact (e.g. object position and shape) is interesting, although not extremely surprising.
- The observed failure of supervised models to generalize to unseen combinations is interesting.

I generally feel that the paper could provide deeper insights into why the observed failure modes occur and how they could be mitigated, specifically:
- The concept of encoder/decoder failure is interesting. However, I feel the paper does not provide a comprehensive investigation of this issue, since the encoder and decoder are trained jointly on data with missing factor combinations. What would the results look like for an encoder/decoder pretrained without missing combinations and then trained together with a randomly initialized decoder/encoder on data with missing combinations?
- What is the role of the prior? The fact that the training data does not contain all combinations of the factors should be reflected in the prior. In the current setup, it is possible that enforcing a jointly Gaussian prior could hurt combinatorial generalization.
- In the supervised setting it would be interesting to see whether rebalancing the training set such that the targets are jointly uniformly distributed improves combinatorial generalization.
- In most of the examples, for ordinal factors an interval of values was left out, including the minimum/maximum value. How does including the minimum and maximum values for ordinal factors affect combinatorial generalization? Does seeing the minimum/maximum value improve combinatorial generalization for combinations of ordinal factors (Section 3.2 shows that this is not the case for categorical factors).

In summary, I think the paper addresses an important problem and explores an interesting direction, but could provide more insights and potentially strategies to mitigate combinatorial generalization failures.


Typos:
- L162 achieve instead of access?
- L196 posX


_Post-rebuttal update_

With the investigations added in response to the review the failure cases are now characterized more in-depth and more comprehensively. I therefore adapted my score. I would suggest to summarize the experiments with rebalanced data, frozen decoder, and those suggested by reviewer HV8h in the main paper, with references to the appendix, to ensure that these aspects get noticed by readers who wish to build on top of this work.

---

> ### Author Response · Authors · 2022-08-02
> **Response to review**
>
> Thank you for a set of really useful suggestions. We have carried out a set of simulations specifically to address the reviewers comments (see General comments above and the revised Supplemental Materials). Here we address some specific comments of the reviewer.
>
> R3.1. Encoder / Decoder failure: Thank you for this suggestion. We really liked the reviewer's suggestion and have carried out something very similar to what the reviewer recommended. Please see point G3 above where we discuss this in more detail.
>
> R3.2. Role of prior: This is a good point. There are two things to say here: Firstly, it is not entirely clear to us how such a prior about unknown parts of generative factors be included in the prior. Secondly, and more importantly, in real-world situations it is not always feasible to encode the biases in the training set in the prior. In many cases, a system does not know which combinations are likely to be left out in advance and it may still need to generalise to unseen combinations. Human beings are particularly good at this type of generalisation and we wanted to investigate why VAEs seem to fail.
>
> R3.3. Rebalancing training set: This is again an excellent suggestion. We have carried out this simulation. See point G2 above where we discuss the results. In short, we observed that rebalancing helped the models increase disentanglement but did not help with generalisation in output of latent space.
>
> R3.4. Including minimum & maximum values: We had already performed this comparison for the Circles dataset in our manuscript (Figure 4 on page 6). Our observation was that the model fails to generalise even when minimum & maximum values are included if there is confounding value in one of the factors that is excluded during training (compare Figures 4.a and 4.b where the addition of shape breaks the model). If the reviewer is keen for us to test this on the other datasets, we can try and implement this by the end of the discussion period.

---

> > ### Comment · Reviewer_DoYc · 2022-08-07
> > **Response to the author response**
> >
> > I thank the authors for their response to my comments as well as the additional experiments. It looks like the generalization failure cases are now explored more in-depth and more comprehensively, so I will consider raising my score accordingly.

---

### Official Review · Reviewer_Tsuf · 2022-07-11

**Rating:** 7
**Confidence:** 4
**Soundness:** 3 good
**Presentation:** 4 excellent
**Contribution:** 3 good

**Summary:**

This paper studies the generalization ability of disentangled representations through a set of empirical results.
Particularly, this paper studies:
- Do the models that disentangle well, fully capture the combinatorial structure of the world? and the answer is probably not!
- So does the encoder fail to correctly map the unseen test examples to latent space or the decoder is the problem? It appears that in cases that the model doesn't generalize, the encoder fails too. So Most probably the encoder's fault.
- Then why does the model learn to properly disentangle on the training set? By associating latent features with perceptual inputs rather than learning how these factors interact with each other.
- Is this problem limited to SSL tasks? No, even in supervised tasks we can see that the encoder fails to map properly to the latent space.
- Then how does it explain other works showing the good performance? They either vary a single factor of variation or factors that do not interact much in the input space.


**Questions:**

I would like to know authors' opinions on how one would go about solving the encoder's lack of generalizability?
It appears that the paper claims that the encoder somehow "overfit" on the training distribution. Or it rather undefits and fail to capture the interactions between the factors of variation?

**Strengths And Weaknesses:**

**Strengths**
- The paper studies a relevant problem, interest of the community.
- It is very well structured. As I read through the paper, some questions raise in my mind and then I find the answer in some later paragraphs.
- The hypothesis, experiment, and then result and intuitions are separated in a clean way. Easy to follow!
- I also enjoyed reading section 4 and the discussion afterwards. Very well concludes the work.

**Weaknesses**
- I detected only some minor errors such as using "it's" instead of "its" and inconsistent citations.

**Disclaimer**
My filed is more generalization rather than disentanglement. Hence, I cannot fully evaluate the significance of the works. From a generalization point of view, I found this work eye-opening and beautifully done. On disentanglement, I am keen to read other reviews on this work.

---

> ### Author Response · Authors · 2022-08-02
> **Response to review**
>
> Thank you for the positive comments. We really appreciate that you related to our work and found it insightful. We have carried out a set of simulations to further test the robustness of the findings (see General Comments above and the revised Supplementary Materials) that the reviewer may be interested in.
>
> R2.1. Typos: Thank you for catching these - we will make these corrections in the camera-ready version.
>
> R2.2. Solving generalisation: We agree, it would be great to propose a solution to this problem. As discussed above (point G5), we don’t believe this will be trivial. We believe that in order to perform combinatorial generalisation, a model will need to solve two different problems (I) abstraction - I.e. inferring the correct invariances (eg. scaling doesn’t change a shape just the area that it covers in an image), and (ii) a specific instance of the binding problem where abstract representations are bound to visual stimuli (e.g., binding the concept of square to locations in the image). One particular approach that holds some promise is object centric models (eg. SlotAttention (Locatello, etal, 2020))l which tries to segment an image into different components. We are in the process of investigating whether such a model can solve the problems we have highlighted in our manuscript.

---

> > ### Comment · Reviewer_Tsuf · 2022-08-02
> > **Thank you**
> >
> > I appreciate that authors' reply!
> > I agree with the authors that "falsification" is an important step in identifying a problem and solving the problem comes at the next step. While respecting other reviewers' opinions, I still believe the community would benefit from publication of this paper.
> >
> > Best of luck!

---

### Official Review · Reviewer_vDTx · 2022-07-11

**Rating:** 7
**Confidence:** 4
**Soundness:** 2 fair
**Presentation:** 2 fair
**Contribution:** 1 poor

**Summary:**

The paper analyses failure cases of deep disentanglement models. It focuses specifically on their ability to deal with combinatorial generalization, i.e. to disentangle at test time attributes that are presented in combinations that have not been observed in the training data.
The paper makes the distinction between encoder errors, corresponding to when the values in the latent variable do not reflect the actual values of the generative attributes, and decoder errors, when the latent code is correctly inferred but the decoder is not able to recombine the values correctly to build the output.

They find, as in previous work, that in the more challenging evaluation conditions, the models fail to disentangle.
They observe that the errors already appear in the latent code (encoder errors) and that some attributes (especially "shape") were more difficult to combine than others.
Those observations seem to carry across different settings: VAE and WAE backbones, with or without SBD in the architecture, and in both their unsupervised and supervised trainings.

**Questions:**

- Regarding non-identifiability (2.1.1): The problem is of course well known and would deserve more attention, but I would point out that 1) in the commonly used settings, many models do seem to work nevertheless 2) it is unclear what in the proposed setup from Montero et al. is dedicated to solving this particular problem. Consequently, I find l.100 a little bit misleading, as it can imply that Motenro et al solve this problem in opposition to other methods that can't. Can the author provide some clarification about what is said in this paragraph?
- Semi-supervised or unsupervised: The setup described in 2.1.1. is referred to as semi-supervised. Indeed, it needs supervision in the sense that labels are necessary to be able to present the correct output given the three inputs. However, in the rest of the paper, there is no mention of the semi-supervised setting. Instead, it seems to be refered to as unsupervised. Moreover, in the supplementary, the authors are mentionning "weak supervision signal" to explain the same setting.
- Following the previous issue, if the presented setting is semi-supervised, then I think the paper also needs to add experiments in a fully unsupervised setting. Indeed, the studied methods derive from the unsupervised disentanglement literature and are aimed to address the specific challenges of this difficult setting. There is value in relaxing some constraints for a study, but it remains necessary to check if the findings hold in a fully unsupervised setting. If the authors wanted to fully address the supervised disentanglement method as well, they would have to encompass a much broader scope of related work, including ELEGANT for instance. Can the author clarify the positionning of the paper regarding supervision, as well as to attribute swapping approaches?
- The authors mention in supplementary that they were not able to train MPI3D with a VAE, which is a little bit surprising considering they can be trained on more complex data, and that other works report some success on this dataset with models of the same family. [2] Can the authors comment on what in there setting makes it impossible?
- In the Discussion section, the author attempt to explain their results with the concept of "interaction" that is not defined. They state that shape and color for instance are attributes that interact together, while color and position do not. But of course, it could be argued that position itself is composed of two variables that interact to produce the output. What about the orientation attribute, should it be considered as interacting with shape? with position? Can the authors explicit what they mean by interacting attributes? and provide some results with the other attributes to make the generalization somewhat more grounded? Currently, it only considers one attribute per dataset, which is not enough to draw any hypothesis. The discussion seems to be very weak to me.

Mainly, the writing of the paper lacks precision.
I would consider raising the rating if the authors can address this issue by clarifying the mentioned points and showing convincingly that the camera-ready will be less ambiguous.

[1] ELEGANT: Exchanging Latent Encodings with GAN for Transferring Multiple Face Attributes.
Taihong Xiao, Jiapeng Hong, Jinwen Ma. ECCV 2018.

[2] Weakly-Supervised Disentanglement Without Compromises.
Francesco Locatello, Ben Poole, Gunnar Rätsch, Bernhard Schölkopf, Olivier Bachem, Michael Tschannen. ICML 2020

**Limitations:**

The authors do not adress the limitations. They should maybe mention the limited complexity of the data the presented models are able to fit, as it is a general limitation of the field.

---

Post-Rebuttal: My main concerns have been addressed in the revised submission. I now recommend acceptance, and accordingly, change my rating from 3 to 7.

---

**Strengths And Weaknesses:**

Strengths:
- The paper tackles the challenging task of analyzing and studying a not well-understood behavior of generative models (disentanglement).
- The authors try a large combination of settings and make observations that seem to generalize across the settings.

Weaknesses:
- There seems to be no strong new idea, the model, architecture, training, metrics, and experimental setups are all taken from prior work. But the paper is framed as a deeper study of some preliminary observations made in previous works, so it's ok.
- The paper could be presented more clearly. Some important descriptions are ambiguous.
- The observations are interesting but the link between the evidence and the interpretations drawn from the authors seems very weak.

---

> ### Author Response · Authors · 2022-08-02
> **Response to review**
>
> Thank you for a set of really useful comments and suggestions. We have made several changes to the manuscript and added simulations to a revised Supplementary Material (Appendices C & D). We have responded to the most important comments above under General Responses and here we respond to some specific comments made by the reviewer.
>
> R1.1. Non-identifiability: We appreciate that the wording used in section 2.1.1 might have been misleading. The claim is not that Montero et al. solves this problem and other approaches don`t, but that this task is similar to other successful approaches. Indeed, as we state in the appendix (Section A.1), the task works based on its similarity to other approaches (with weak-supervision or sparse transitions between factors) that consistently induce higher levels of disentanglement. Nevertheless, to make it clear, we have replaced the wording in Section 2.1.1 to state that `` To get around this problem we used the composition task developed by Montero et al. [25] (see Appendix A.1 for further discussion)”.
>
> R1.2. Semi-supervised or Unsupervised: Thank you for catching these inconsistencies. We have now revised the manuscript to ensure that we use semi-supervised consistently in the manuscript. As far as we can test, the results also seem to apply to the unsupervised setting (see point G4 above), but we agree that more investigation is needed to confirm this for all cases. We also want to clarify here that once the model has been trained on the semi-supervised setting, all our tests are done for the unsupervised (image-reconstruction) task.
>
> R1.3. Fully unsupervised and supervised settings: We agree that the unsupervised setting should be checked and we have been able to do this to a certain extent using a new method (see point G4 above). Regarding the supervised setting, it wasn't our intention to argue that our results generalise to all supervised settings. What we intended to do was to address the question whether further disentanglement (beyond the best observed in current models) can help with generalisation. Therefore, we used a setting where the encoder is trained with ground-truth latent values, which are completely disentangled. We called this condition the supervised setting as it mirrors a neural network mapping images to a set of labels. In hindsight, this has led to more confusion. Therefore, we have changed the name of this setting to "ground-truth latents", which seems more intuitive.
>
> R1.4. Training MPI3D:
> We first trained using the same parameters found in the original MPI3D paper (Gondal et al., 2019) but we did not manage sharp enough reconstructions. Note that even in their article reconstructions appear blurry upon closer inspection. This makes it difficult to establish if the models have been successfully trained or not. We attempted with larger architectures, which did indeed produce better reconstructions but at the expense of longer training times. We thus decided to test WAEs as they report better learning performance than traditional VAEs. This enabled us to train the models at a much faster pace and does not undermine our results since we achieve the same levels of disentanglement as before.
>
> R1.5. Discussion of Interaction: Thank you for pointing this out. We agree that due to the space limitations, we were not able to go into any detail about what our hypothesis about interaction is. Based on your feedback, we have now written a section in the Appendix specifically to clarify what we mean by “interactive” vs “non-interactive” conditions (Appendix C and Figure 29). The reviewer is right that indeed `shape’ interacts with not just ‘object-hue’ in 3DShapes dataset, but it will also interact with orientation (aka camera angle). And indeed, in dSprites, shape interacts with position, orientation as well as scale. The only example of non-interactive condition that we could study in these datasets was the floor-hue vs wall-hue and, as we discuss in the manuscript, the model succeeds in these non-interactive conditions. If the reviewer finds our discussion (and Figure) about clarifying interactive conditions useful, we will be happy to move it to the main text in the camera-ready version.

---

> > ### Comment · Reviewer_vDTx · 2022-08-05
> > **Discussion**
> >
> > Thank you for the rebuttal and the revisions to the submission. The setup described in G4 is a nice addition.
> >
> > I also appreciate the clarification about *interactions*, the argument becomes easier to get behind. Still, it's worth pointing out that not all interacting features seem to be equal and there is probably more to the story. Figure 13 and 15 indicate that posX and posY seem to generalize in the center condition but not in the corner condition, while shape doesn't generalize in either.
> > There is also a broken reference in section C of the supplementary.
> >
> > Overall, I believe they addressed my most pressing concerns about clarity and precision, and I will revise my ratings accordingly.

---

> > > ### Author Response · Authors · 2022-08-06
> > > **Discussion**
> > >
> > > Thank you for your feedback. We are really pleased to hear that we were able to address your most important concerns.
> > >
> > > We completely agree with your point about “not all interactions are equal”. For example, on the graphical view (Figure 29) there may be some interactions – such as when the pixel value is a linear combination of the two generative factors – where the model may be able to learn the dependencies between the pixels and generative factors. We will be happy to add this clarification to the camera-ready version.

---

### Author Response · Authors · 2022-08-02
**General comments part I**

We thank the reviewers for their feedback and very useful suggestions. We respond to the key comments here and some specific comments below to each reviewer.

In response to these concerns, we have carried out an additional set of simulations and modifications to the manuscript. Here are the highlights:

G1. Based on the suggestions of Reviewer HV8h, we have run a subset of our experiments on CascadeVAE. As argued by the reviewer, this model represents some latent variables as discrete values, so it is interesting to check whether our results generalise to these set of models. We show that the model learns the task but fails to achieve combinatorial generalisation. Due to the discrete nature of latents in CascadeVAE, the failures are even more stark, and one can clearly see how the model swaps a novel value of a generative factor (e.g., shape=square) with a value seen during training (e.g., shape=heart). We have added the results for an example model in the Appendix D, Figure 35 and we will be happy to move this to the main text for the camera-ready version, if the reviewers feel this will be useful.


G2. Reviewer DoYc suggested running a rebalanced dataset, where the different values of the shape are uniformly sampled. (In the submitted manuscript, the variables with the left-out combination had fewer samples in the training dataset). We have carried out simulations for a subset of experiments and our observation is that rebalancing increases the degree of disentanglement. But critically, it does not help with generalisation (see new Figure 31 in Appendix D). In fact, with the increased level of disentanglement, we are more clearly able to see the failures both in the reconstructions and in the latent space.

G2.1 The reviewer also suggested excluding items from the training set which were in the middle of the range. We already did this for the Circles dataset (Figure 4) and found that the model fails to generalise as long as enough training samples are excluded (reconstruction-to-range condition).


G3. Reviewer DoYc suggested that a potential problem could be the joint training of the encoder and decoder and to properly investigate whether generalisation fails in the latent space would be to use a pre-trained encoder/decoder without missing combinations and then combine them. We were also concerned about this issue of joint training of the encoder and decoder, and this was the motivation behind running the supervised task (aka latent prediction task in Section 3), where we trained only the encoder and found that even having ground-truth latents as targets did not help with generalisation.
Nevertheless, we really liked the reviewer's suggestion as this allows us to verify our results using a different method.  Following the reviewer’s suggestion, we created a “Frankenstein” model by replacing the decoder of an untrained model with a decoder that had been pre-trained on the full dataset (and the weights of the decoder frozen). Then the complete VAE (with a new encoder) was trained end-to-end on the (unsupervised) dSprites image reconstruction task. We verified that the decoder learned highly disentangled representations (Figure 34.a) and tested this Frankenstein model on the trained as well as held-out combinations. The results of this simulation have been added to Appendix D.2 (Figure 34.b and 34.c). As can be seen, despite having the availability of this "ideal” decoder (which has seen the entire dataset), the encoder still fails to project the left-out combinations in the correct part of the latent space. These results are consistent with other results in the manuscript (compare with Figure 3 in the main text) showing that the failure in generalisation is not limited to the failure of the decoder.

---

> ### Author Response · Authors · 2022-08-02
> **General Comments Part II**
>
> G4. Reviewer vDTx was concerned about how our results generalise from the semi-supervised task to an unsupervised task (i.e. image reconstruction). In our manuscript we have run the simulations on the semi-supervised 'composition task' because this task forces the model to discover highly disentangled representations. These highly disentangled representations are needed to analyse the latent space (it becomes very difficult to analyse whether the model projects to the correct region of latent space for unseen combinations when latent representations are not disentangled). In general, we have found that the latent representations are not easy to interpret for almost all models we tested under the unsupervised learning task. However, the simulations discussed in point G3 above does address the reviewer's concern to some extent. In these simulations, we trained an ideal decoder and therefore we were able to work with highly disentangled representations for an unsupervised learning task. The results of these simulations (see Figure 34 in Appendix D.2) are consistent with the results for the semi-supervised task: the model projects the unseen combinations during the unsupervised learning task to the incorrect region of the latent space. Please also see the results for the new simulations using CascadeVAE (Figure 35 in Appendix D.3) where we again find that the model fails at combinatorial generalisation in the unsupervised learning task.
>
>
> G5.  Reviewers Tsuf and HV8h mentioned that they would like to see how this problem can be solved. We understand that looking for a solution makes sense from an engineering perspective, and we share the reviewer's frustration at not knowing how to solve this problem. However, we believe that the solution to the problem of combinatorial generalisation is not trivial. In order to properly generalise the model needs to not only understand how to generalise for a given set of values on each variable, but also understand how a variable (e.g., orientation, or shape) "works". Now, we are aware that some generalisation solutions have been proposed for some variables (such as orientation) but it remains unclear how a novel shape can be combined with a novel orientation without having a process of simulating the property of rotation on a given shape -- something that humans seem to do (e.g., see Shepard & Metzler, 1971). We also believe that from a scientific point of view, identifying problems with existing models is an essential first step towards solving these issues. In our view, the role of falsification in science has been undervalued in the ML community (incidentally, the topic of an upcoming workshop NeurIPS this year, entitled: “Workshop on Understanding Deep Learning Through Empirical Falsification”)
>
> In addition to these simulations, we have made several modifications to the manuscript to increase clarity (including suggestions made by reviewer vDTx) and responded to each reviewer's comments below. We will love to get the reviewer's thoughts on these responses and changes.

---

### Author Response · Authors · 2022-08-08
**Post-rebuttal comments**

We thank the reviewers for a constructive rebuttal period and we are pleased that we have managed to address their concerns.

We have added a substantial amount of extra simulations in the appendix as well as two extra models (CascadeVAE and LieGroupVAE) which address several concerns (lack of discrete representations, no explicit modeling of interactions and unsupervised training). We have been unable to make any references to these additions in the main text due to lack of space, but if accepted we will include them in the main section at the reviewers request.

---

### Meta-Review · Area_Chair_H7yd · 2022-08-22

**Recommendation:** Accept
**Confidence:** Certain

**Metareview:**

All reviewers acknowledge that the paper conducts an extensive study of the combinatorial generalization of models, and the additional experiments on CascadeVAE and LieGroupVAE conducted by the authors convinced the reviewers. Despite the lack of a proposed solution, I believe this paper provides novel interesting insights on the failure modes of disentanglement models (for example, the fact that if the factor affect different part of the image combinatorial generalisation is much more likely to occur). In agreement with the reviewers, we recommend acceptance.



**Award:**

No

---

### Decision · Program_Chairs · 2022-09-14

Accept